# THE GEOMETRY OF REASONING: FLOWING LOGICS IN REPRESENTATION SPACE

**Yufa Zhou**[*], **Yixiao Wang**[*], **Xunjian Yin**[*], **Shuyan Zhou, Anru R. Zhang**
Duke University
{yufa.zhou,yixiao.wang,xunjian.yin,shuyan.zhou,anru.zhang}@duke.edu

## ABSTRACT

We study how large language models (LLMs) "think" through their representation space. We propose a novel geometric framework that models an LLM's reasoning as flows—embedding trajectories evolving where logic goes. We disentangle logical structure from semantics by employing the same natural deduction propositions with varied semantic carriers, allowing us to test whether LLMs internalize logic beyond surface form. This perspective connects reasoning with geometric quantities such as position, velocity, and curvature, enabling formal analysis in representation and concept spaces. Our theory establishes: (1) LLM reasoning corresponds to smooth flows in representation space, and (2) logical statements act as local controllers of these flows' velocities. Using learned representation proxies, we design controlled experiments to visualize and quantify reasoning flows, providing empirical validation of our theoretical framework. Our findings indicate that training solely via next-token prediction can lead LLMs to internalize logical invariants as higher-order geometry in representation space, challenging the "stochastic parrot" argument. Experiments across Qwen and LLaMA model families further suggest the presence of a general, possibly universal, representational law underlying machine understanding and human linguistic regularities, largely independent of specific training recipes or model architectures. Our work serves as both a conceptual foundation and practical tools for studying reasoning phenomena, offering a new lens for interpretability and formal analysis of LLMs' behavior.

🐙 **Code:** https://github.com/MasterZhou1/Reasoning-Flow
🤗 **Dataset:** https://huggingface.co/datasets/MasterZhou/Reasoning-Flow

> "Reasoning is nothing but reckoning."
> — *Thomas Hobbes*

## 1 INTRODUCTION

The geometry of concept space, i.e., the idea that meaning can be represented as positions in a structured geometric space, has long served as a unifying perspective across AI, cognitive science, and linguistic philosophy (Gardenfors, 2004; Rickard, 2006; Gardenfors, 2014). Early work in this tradition was limited by the absence of precise and scalable semantic representations. With the rise of large language models (LLMs) (Hurst et al., 2024; OpenAI, 2025; Grattafiori et al., 2024; Guo et al., 2025; Yang et al., 2025), we revisit this geometric lens: pretrained embeddings now offer high-dimensional vector representations of words, sentences, and concepts (Mikolov et al., 2013a;b; Arora et al., 2018; Neelakantan et al., 2022; Zhang et al., 2025a; Lee et al., 2025; Kozlowski et al., 2025), enabling geometric analysis of semantic and cognitive phenomena at scale.

A seminal recent work (Modell et al., 2025) formalizes the notion that learned representations in LLMs lie on low-dimensional concept manifolds. Building on this view, we hypothesize that

---

[*]Equal contribution.

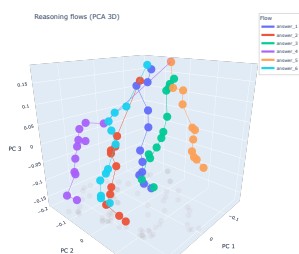 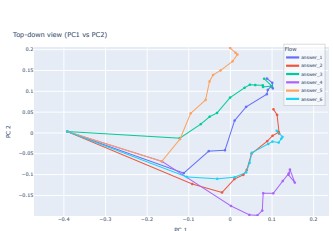 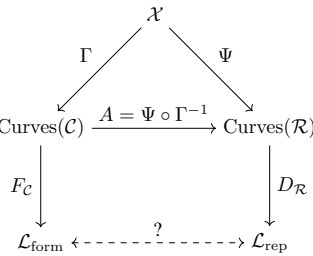

(a) Reasoning flows visualized using PCA in 3 dimensions.

(b) Reasoning flows visualized using PCA in 2 dimensions.

(c) Schematic illustration of mappings between spaces.

Figure 1: Reasoning Flow. (a–b) Visualizations on a selected problem from MATH500 with six distinct answers. (c) Our geometric framework of mapping relationships among input space $\mathcal{X}$, concept space $\mathcal{C}$, logic space $\mathcal{L}$, and representation space $\mathcal{R}$. See Section 4 for more details.

reasoning unfolds as a trajectory, potentially a flow, along such manifolds. To explore this idea, we draw on classical tools from differential geometry (Menger, 1928; Hicks, 1965; Guggenheimer, 2012; Do Carmo, 2016) and propose a novel geometric framework for analyzing reasoning dynamics in language models. Concretely, we view reasoning as a context-cumulative trajectory in embedding space: at each step, the reasoning prefix is extended, and the model's representation is recorded to trace the evolving flow (Figures 1a and 1b). Our results suggest that LLM reasoning is not merely a random walk on graphs (Wang et al., 2024a; Minegishi et al., 2025). At the isolated embedding level, trajectories exhibit stochasticity reminiscent of graph-based views; however, when viewed cumulatively, a structured flow emerges on a low-dimensional concept manifold, where local velocities are governed by logical operations. To the best of our knowledge, this is the first work to formalize and empirically validate such a dynamical perspective, offering quantitative evidence together with broad insights and implications. We further rigorously define and formalize concept, logic, and representation spaces (Figure 1c), and relate them through carefully designed experiments. See the full theoretical definitions of flow, velocity, curvature, and the associated spaces in Section 4.

From Aristotle's syllogistics to Frege's predicate calculus and modern math foundations (Bochenski & Thomas, 1961; Copi et al., 2016; Enderton, 2001), formal logic isolates validity as form independent of content. Wittgenstein's *Tractatus* sharpened this view—"the world is the totality of facts, not of things" (Wittgenstein, 1922)—underscoring logical form as the substrate of language and reality. In this spirit, we treat logic as a carrier-invariant skeleton of reasoning and test whether LLMs, trained on massive corpora, have internalized such structural invariants on the embedding manifold, effectively rediscovering in data the universal logic that took humans two millennia to formalize. We deliberately construct a dataset that isolates formal logic from its semantic carriers (e.g., topics and languages) to validate our geometric perspective.

Our experiments, conducted with Qwen3 (Yang et al., 2025) hidden states across model families on our newly constructed dataset, reveal that LLMs exhibit structured logical behavior. In the original (0-order) representation space, semantic properties dominate, with sentences on the same topic clustering together. However, when we analyze differences (1- and 2-order representations), logical structure emerges as the dominant factor. Specifically, we find that velocity similarity and Menger curvature similarity remain highly consistent between flows sharing the same logical skeleton, even across unrelated topics and languages. In contrast, flows with different logical structures exhibit lower similarity, even when they share the same semantic carrier. Our further random shuffle experiment shows that permuting the order of logical statements collapses velocity and curvature yet preserving positional similarity, indicating that logical structure is encoded in higher-order geometry rather than in raw representation space. These findings provide quantifiable evidence supporting our hypothesis that logic governs the velocity of reasoning flows, challenging the "stochastic parrot" view (Bender et al., 2021) that LLMs lack genuine understanding. Our results suggest a general, possibly universal, constraint on language representations, aligning with the Platonic Representation Hypothesis (Huh et al., 2024; Jha et al., 2025), which posits that neural networks can learn shared underlying world representations largely independent of architecture and training data.

While interpretability research on LLMs has made substantial empirical progress (Anthropic, 2021; Rai et al., 2024; Nanda et al., 2023; Singh et al., 2024; Madsen et al., 2024; Ferrando et al., 2024), rigorous theoretical understanding remains comparatively limited, with only a few recent efforts in this direction (Jiang et al., 2023; Park et al., 2024b; Modell et al., 2025; Park et al., 2025). Our work contributes to this emerging line by introducing a mathematically grounded framework with formal definitions and analytic tools for quantifying and analyzing how LLMs behave and reason. We hope our theory and empirical evidence open a new perspective for interpretability community and spark practical applications. Our contributions are:

- We introduce a geometric perspective that models LLM reasoning as flows, providing formal definitions and analytic tools to study reasoning dynamics.

- We design a formal logic dataset that disentangles logical structure from semantic surface, enabling direct tests of whether LLMs internalize logic beyond semantics.

- We empirically validate our framework through experiments and analysis, demonstrating its utility and offering practical insights.

## 2 RELATED WORK

**Concept Space Geometry.** The Linear Representation Hypothesis (LRH) proposes that concepts align with linear directions in embedding space, a view supported by theoretical analyses and empirically validated in categorical, hierarchical, and truth–false settings (Park et al., 2024b; Jiang et al., 2024; Park et al., 2025; Jiang et al., 2023; Marks & Tegmark, 2024). However, strict linearity is limited: features may be multi-dimensional or manifold-like, as seen in concepts like colors, years, dates, and antonym pairs. (Engels et al., 2025; Modell et al., 2025; Kozlowski et al., 2025). Other works emphasize compositionality, showing that concepts require explicit constraints or algebraic subspace operations to compose meaningfully (Stein et al., 2024; Wang et al., 2023). At a broader scale, hidden-state geometry follows expansion–contraction patterns across layers and exhibits training trajectories whose sharp shifts coincide with emergent capabilities and grokking (Valeriani et al., 2023; Park et al., 2024a; Liu et al., 2022). Sparse autoencoders further reveal multi-scale structure, from analogy-like "crystals" to anisotropic spectra (Li et al., 2025b). Collectively, these results suggest that concept spaces are locally linear yet globally curved, compositional, and dynamic, motivating our perspective of reasoning as flows on such manifolds.

**Understanding Reasoning Phenomena.** LLMs benefit from *test-time scaling*, where allocating more inference compute boosts accuracy on hard tasks (Snell et al., 2025). Explanations span expressivity—CoT enabling serial computation (Li et al., 2024), reasoning as superposed trajectories (Zhu et al., 2025), and hidden planning in scratch-trained math models (Ye et al., 2025)—to inductive biases, where small initialization favors deeper chains (Yao et al., 2025). Structural analyses view reasoning as path aggregation or graph dynamics with small-world properties (Wang et al., 2024a; Minegishi et al., 2025), while attribution highlights key "thought anchors" (Bogdan et al., 2025). Empirical work shows inverted-U performance with CoT length and quantifiable reasoning boundaries (Wu et al., 2025; Chen et al., 2024), and embedding-trajectory geometry supports OOD detection (Wang et al., 2024b). Cai et al. (2025) shows RL is more sensitive to backtracking structure than content correctness, clarifying SFT–RL interplay in reasoning. Moving beyond text, latent-reasoning methods scale compute through recurrent depth, continuous "soft thinking," and latent CoT for branch exploration and self-evaluation (Zhang et al., 2025b; Geiping et al., 2025; Hao et al., 2024; Wang et al., 2025). Applications exploit these insights for steering and efficiency: steering vectors and calibration shape thought processes (Venhoff et al., 2025; Chen et al., 2025), manifold steering mitigates overthinking (Huang et al., 2025), and adaptive indices enable early exit (Fu et al., 2024).

## 3 PRELIMINARIES

### 3.1 LARGE LANGUAGE MODELS

Let $\mathcal{V}$ denote a finite vocabulary of tokens, and let $\theta$ denote the parameters of a large language model (LLM). An LLM defines a conditional probability distribution $p_\theta(u_t \mid u_{<t}, P), \quad u_t \in \mathcal{V}$, where

$u_{<t} := (u_1, \ldots, u_{t-1})$ is the prefix of previously generated tokens and $P \in \mathcal{V}^n$ is the tokenized problem prompt. At each step $t$, inference proceeds by sampling $u_t \sim p_\theta(\cdot \mid u_{<t}, P)$.

**Definition 3.1** (Chain-of-Thought Reasoning). *Given a prompt $P \in \mathcal{V}^n$, Chain-of-Thought (CoT) reasoning is an iterative stochastic process that generates a sequence $\mathcal{U} = (u_1, u_2, \ldots, u_T)$, $u_t \in \mathcal{V}$, via recursive sampling $u_t \sim p_\theta(\cdot \mid P, u_{<t})$, $t = 1, \ldots, T$.*

To enable geometric analysis of reasoning, we need a mapping from discrete token sequences into continuous vectors, a transformation that modern LLMs naturally provide.

**Definition 3.2** (Representation Operator). *A Representation Operator is a mapping $\mathcal{E} : \mathcal{V}^* \times \mathcal{I} \to \mathbb{R}^d$, where $x = (x_1, \ldots, x_n) \in \mathcal{V}^*$ is a token sequence and $\iota \in \mathcal{I}$ is an index specifying the representation type (e.g., a token position, a pooling rule, or an internal layer state). The output $\mathcal{E}(x, \iota) \in \mathbb{R}^d$ is the embedding/representation of $x$ under the selection rule $\iota$. For notational simplicity, we omit the index $\iota$ unless explicitly required.*

The range of this operator defines the ambient space of reasoning:

**Definition 3.3** (Representation Space). *Given a representation operator $\mathcal{E}$, the representation space is $\mathcal{R} := \{\mathcal{E}(x) : x \in \mathcal{V}^*\} \subseteq \mathbb{R}^d$. Elements of $\mathcal{R}$ are continuous embeddings of discrete language inputs, serving as the foundation and empirical proxy for our geometric analysis of reasoning.*

In practice, $\mathcal{E}$ may be instantiated by a pretrained encoder such as Qwen3 Embedding (Zhang et al., 2025a) or OpenAI's `text-embedding-3-large` (Neelakantan et al., 2022), or by extracting hidden states directly from an LLM. Typical choices of $\iota$ include mean pooling, the hidden state of the final token, or a specific layer–position pair within the model (Zhang et al., 2025a; Lee et al., 2025; Hessani, 2025; Nie et al., 2024). We interpret $\mathcal{E}$ as projecting discrete language sequences into a continuous semantic space, potentially lying on a low-dimensional manifold embedded in $\mathbb{R}^d$ (Modell et al., 2025; Engels et al., 2025; Kozlowski et al., 2025).

### 3.2 MENGER CURVATURE

We adopt *Menger curvature* (Menger, 1928) to quantitatively capture the geometric structure of reasoning flows. As a metric-based notion of curvature, Menger curvature simultaneously reflects both angular deviation and distance variation, making it particularly suitable for reasoning trajectories represented as discrete embeddings. We leave more details to Appendix D.2.

**Definition 3.4** (Menger Curvature). *Let $x_1, x_2, x_3 \in \mathbb{R}^n$ be three distinct points. The Menger curvature of the triple $(x_1, x_2, x_3)$ is defined as the reciprocal of the radius $R(x_1, x_2, x_3)$ of the unique circle passing through the three points: $c(x_1, x_2, x_3) = \frac{1}{R(x_1, x_2, x_3)}$.*

## 4 REASONING AS GEOMETRIC FLOWS IN REPRESENTATION SPACE

We formalize the view that LLMs reason by tracing trajectories in their representation space. A central question is whether LLMs exhibit intrinsic control over these flows, mirroring the human perspective. We hypothesize semantic content as a curve on a concept manifold, and logical structure acts as a local controller of the trajectory. In this section, we introduce the spaces, maps, and geometric quantities that underpin the paper. We then rigorously formalize this construction and establish the correspondence between the LLM's representation space and the human concept space.

### 4.1 CONCEPT SPACE AND SEMANTIC TRAJECTORIES

**Definition 4.1** (Concept Space). *The concept space $\mathcal{C}$ is an abstract semantic space that models human-level cognitive structures such as ideas, reasoning states, and problem-solving subtasks.*

We assume $\mathcal{C}$ is endowed with a smooth geometric structure, allowing continuous trajectories to represent the evolution of conceptual content. This assumption can be traced back to the classical insight of William James (James et al., 1890), who famously argued that consciousness does not appear to itself "chopped up in bits." Chains or trains of thought are, in his words, inadequate metaphors; instead, "it is nothing jointed; it flows. A river or a stream are the metaphors by which it is most naturally described."

**Definition 4.2** (Semantic Subspace as Cognitive Trajectories). *Let $\mathcal{M} \subseteq \mathcal{C}$ denote a semantic subspace corresponding to a coherent domain of meaning (e.g., temporal concepts, colors, or causal relations). Let $\mathcal{X}^*$ denote the set of all finite input sequences over $\mathcal{X}$. We introduce a* trajectory map

$$\Gamma : \ \mathcal{X}^* \to \mathrm{Curves}(\mathcal{M}), \qquad X \mapsto \gamma_X,$$

*that assigns each sentence $X_T = (x_1, \ldots, x_T)$ to a continuous curve $\gamma_X$ within $\mathcal{M}$. Formally,*

$$\gamma_X : [0,1] \to \mathcal{M}, \qquad s \mapsto \gamma_X(s),$$

*where $s \in [0,1]$ is a continuous progress parameter along the reasoning flow. For each discrete prefix $(x_1, \ldots, x_t)$, we align it with the point $\gamma_X\left(\frac{t}{T}\right)$ on the curve. The curve $\gamma_X$ thus traces the gradual unfolding of semantic content, formalizing the view that human cognition operates as a* continuous flow *of concepts rather than as a sequence of isolated symbols.*

We then define the logic space that mirrors the human view of logic.

**Definition 4.3** (Formal Logical Space). *The formal logical space $\mathcal{L}$ is an abstract domain that captures structural dynamics of reasoning (natural deduction ([Troelstra & Schwichtenberg, 2000](); [Pelletier & Hazen, 2024](#)); see Definition 5.1). Define the flow operator*

$$F_{\mathcal{C}} : \ \mathrm{Curves}(\mathcal{C}) \ \to \ \mathcal{L}_{\mathrm{form}},$$

*which maps a semantic trajectory to its formal counterpart. Semantically different expressions that correspond to the same natural-deduction proposition map to the same element in $\mathcal{L}_{\mathrm{form}}$.*

## 4.2 REPRESENTATION SPACE

We use LLM representations/embeddings as proxies to study human cognition and to investigate why LLMs exhibit reasoning phenomenon. We build on the multidimensional linear representation hypothesis ([Modell et al., 2025]()), which posits that representations decompose linearly into a superposition of features. Each feature corresponds to a basis direction within a feature-specific subspace of the embedding space, weighted by a non-negative activation coefficient encoding its salience.

**Hypothesis 4.4** (Multidimensional Linear Representation Hypothesis ([Modell et al., 2025]())). *Let $\mathcal{X}$ denote the input space (e.g., natural language sentences). Let $\mathcal{F}$ be a set of semantic features. For each feature $f \in \mathcal{F}$, let $\mathcal{W}_f \subseteq \mathbb{R}^d$ denote a feature-specific subspace of the embedding space.*

*Then the representation map $\Psi : \mathcal{X} \to \mathbb{R}^d$ of an input $x \in \mathcal{X}$ is assumed to take the form*

$$\Psi(x) = \sum_{f \in F(x)} \rho_f(x) w_f(x),$$

*where $F(x) = \{f \in \mathcal{F} : \rho_f(x) > 0\}$ is the set of active features in $x$, $\rho_f(x) \in \mathbb{R}_{\geq 0}$ is a non-negative scaling coefficient encoding the intensity or salience of feature $f$ in $x$, $w_f(x) \in \mathcal{W}_f$ is a unit vector ($\|w_f(x)\|_2 = 1$) specifying the direction of feature $f$ within its subspace $\mathcal{W}_f$.*

Building on this compositional picture, we now move from single inputs to *growing contexts*. As a model reasons, its internal representation evolves. The next definition formalizes this evolution as a cumulative flow in embedding space.

**Definition 4.5** (Reasoning Trajectory / Context Cumulative Flow). *Let $\mathcal{X}$ be the input space, and $\Psi : \mathcal{X} \to \mathbb{R}^d$ the representation map from finite input sequences to the embedding space defined in Hypothesis 4.4. Given a prompt $P \in \mathcal{X}$ and a Chain-of-Thought sequence $X_T = (x_1, \ldots, x_T)$ with $x_t \in \mathcal{X}$, define*

$$S_t := (P, x_1, \ldots, x_t), \qquad \widetilde{y}_t := \Psi(S_t) \in \mathbb{R}^d, \qquad t = 1, \ldots, T.$$

*When focusing solely on the reasoning process (ignoring the prompt), we set*

$$y_t := \Psi(X_t) \in \mathbb{R}^d, \quad t = 1, \ldots, T.$$

*The sequence $Y = [y_1, \ldots, y_T] \in \mathbb{R}^{d \times T}$ is called the* context cumulative flow. *The construction of $Y$ follows Algorithm 1.*

---

**Algorithm 1:** Get Context Cumulative Reasoning Trajectory

$\mathcal{V}$: vocabulary space; $P \in \mathcal{V}^n$: tokenized problem prompt; $T$: number of reasoning steps; $x_t \in \mathcal{V}^*$: tokens for step $t$; $\mathcal{E} : \mathcal{V}^* \to \mathbb{R}^d$: representation operator; $y_t \in \mathbb{R}^d$: embedding at step $t$.

---

**Input:** $P \in \mathcal{V}^n$; $X = [x_1, \ldots, x_T]$ with $x_t \in \mathcal{V}^*$
**Output:** $Y = [y_1, \ldots, y_T] \in \mathbb{R}^{d \times T}$
$Y \leftarrow [\,]$, $\quad S_0 \leftarrow [P]$;
**for** $t \leftarrow 1$ **to** $T$ **do**
  $\quad S_t \leftarrow \text{Concat}(S_{t-1}, x_t)$; $\qquad$ // Concatenate with previous context
  $\quad y_t \leftarrow \mathcal{E}(S_t)$; $\qquad\qquad\qquad$ // Get embedding of current step
  $\quad$ Append $y_t$ to $Y$;
**return** $Y$;

---

The embeddings we observe along a sentence are discrete, while reasoning itself is naturally understood to unfold as a continuous process. It is therefore natural to posit an underlying smooth curve from which these discrete points arise as samples, thereby enabling the use of geometric tools such as velocity and curvature.

**Hypothesis 4.6** (Smooth Representation Trajectory). *The discrete representations $\{y_t\}_{t=1}^T$ produced by context accumulation* intrinsically lie *on a $C^1$ curve $\widetilde{\Psi} : [0, 1] \to \mathbb{R}^d$ satisfying*

$$\widetilde{\Psi}(s_t) = y_t \qquad \text{for an increasing schedule } s_1 < \cdots < s_T.$$

In other words, the sequence is not merely fitted by a smooth curve, but should be regarded as samples from an underlying smooth trajectory. This assumption is reasonable: in Appendix D.1 we show an explicit construction of such a $C^1$ trajectory via a relaxed prefix-mask mechanism.

Once a smooth trajectory exists, we can canonically align symbolic progress (e.g., "how far along the derivation we are") with geometric progress in representation space. The following corollary formalizes this alignment on domains where the symbolic schedule is well-behaved.

**Corollary 4.7** (Canonical Alignment). *On a domain where $\Gamma$ is injective and $\widetilde{\Psi}$ is defined, there exists a canonical alignment*

$$A : \text{Curves}(\mathcal{C}) \to \text{Curves}(\mathcal{R}), \qquad A := \widetilde{\Psi} \circ \Gamma^{-1}.$$

**Remark 4.8** (Injectivity of $\Gamma$). *In Corollary 4.7, $\Gamma : \mathcal{X} \to \text{Curves}(\mathcal{C})$ (Definition 4.2) maps linguistic inputs to conceptual trajectories. We do not assume that $\Gamma$ is globally injective over all natural language, as different surface forms may express essentially the same conceptual content. For the purpose of defining the alignment map $A = \widetilde{\Psi} \circ \Gamma^{-1}$, it suffices that $\Gamma$ be injective on a* restricted semantic domain *(e.g., equivalence classes of paraphrases). Understanding global injectivity of $\Gamma$ remains a broader open problem in AI, semantics, and cognitive science research.*

### 4.3 LOGIC AS DIFFERENTIAL CONSTRAINTS ON FLOW

We now turn from the structural hypotheses of representation trajectories to their *dynamical regulation*. In particular, we view logic not as an external add-on, but as a set of differential constraints shaping how embeddings evolve step by step. This perspective enables us to couple discrete reasoning structure with continuous semantic motion.

**Definition 4.9** (Representation-Logic Space). *Given a representation trajectory $Y = (y_1, \ldots, y_T)$ defined in Definition 4.5, define local increments $\Delta y_t := y_t - y_{t-1}$ for $t \geq 2$. The* representation-logic space *is*

$$\mathcal{L}_{\text{rep}} := \{ (\Delta y_2, \ldots, \Delta y_T) \mid Y \text{ a context-cumulative trajectory} \}.$$

The above constructs a discrete object: a sequence of increments capturing how representations change from one reasoning step to the next. To connect this discrete view with a continuous account of semantic evolution, we next introduce the notion of velocity along embedding trajectories.

**Definition 4.10** (Flow Velocity). *Let $\widetilde{\Psi} : [0, 1] \to \mathbb{R}^d$ be the continuous embedding trajectory associated with a sentence. The* flow velocity *at progress $s$ is defined as $v(s) = \frac{d}{ds}\widetilde{\Psi}(s)$, which captures the instantaneous rate of change of the embedding w.r.t. the unfolding of the sentence.*

By relating local increments in representation space (Definition 3.3) to the derivative of a continuous trajectory, we can interpret each discrete reasoning step as an integrated outcome of infinitesimal semantic motion.

**Proposition 4.11** (Logic as Integrated Thought). *By the fundamental theorem of calculus, the cumulative semantic shift between two successive reasoning steps $s_t$ and $s_{t+1}$ is*

$$\int_{s_t}^{s_{t+1}} v(s)\,\mathrm{d}s \;=\; \widetilde{\Psi}(s_{t+1}) - \widetilde{\Psi}(s_t) \;=\; y_{t+1} - y_t \;=\; \Delta y_{t+1}.$$

*Thus, we could view each representation–logic step as the* integration of local semantic velocity*, which aggregates infinitesimal variations of meaning into a discrete reasoning transition. Definition 4.10 captures the central principle that semantic representations evolve continuously, whereas logical steps are inherently discrete: logic acts as the* controller *of semantic velocity, governing both its magnitude and its direction.*

**Continuous–Discrete Correspondence as Structural Constraint.** Having established the conceptual implication of continuous–discrete correspondence, we emphasize that this correspondence reflects the *structural influence of logic on flow dynamics*, rather than a direct mapping from specific inference rules (e.g., conjunction, negation) to vector-field operators.

**Invariance Under Semantic Variation.** We now ask: *what properties of reasoning flows should persist under changes in surface semantics?* We posit that reasoning instances sharing the same natural-deduction skeleton, yet differing in semantic carriers (e.g., topics or languages), should yield reasoning flows whose trajectories exhibit highly correlated curvature (Definition 3.4). If logic governs flow velocity—both magnitude and direction—then flows instantiated with different carriers may undergo translations or rotations, reflecting dominant semantic components of the underlying space. Nevertheless, their *overall curvature should remain invariant*. See a more detailed discussion of curvature in Appendix D.2. Such correlation would indicate that the accumulation of semantic variation produces turning points aligned with both LLM reasoning and human logical thought.

**Bridging Formal and Representational Logic.** This theory directly corresponds to the central objective of this paper: clarifying the relationship between the two logical spaces $\mathcal{L}_{\mathrm{form}}$ and $\mathcal{L}_{\mathrm{rep}}$, as illustrated in Figure 1c. Empirical evidence for this claim will be provided later, where we demonstrate cross-carrier similarity in both first-order differences and curvature.

**Summary.** In summary, logic functions as the *differential regulator of semantic flow*, discretizing continuous variation into meaningful steps. For clarity and reference, all mappings and derivational relationships introduced in this subsection are systematically summarized in Appendix C.

# 5 FORMAL LOGIC WITH SEMANTIC CARRIERS

## 5.1 LOGIC AND NATURAL DEDUCTION SYSTEM

We construct a dataset of reasoning tasks that instantiate the fundamental logical patterns formalized in Definition 5.1. Each task is presented step by step in both formal symbolic notation and natural language. To test whether reasoning relies on surface content or underlying structure, we express the same logical skeletons across diverse *carriers*, e.g., topics such as weather, education, and sports, as well as multiple languages (en, zh, de, ja). This design disentangles logics from linguistic surface and provides a controlled setting for analyzing how reasoning flows behave under varying contexts. Specifically, because all variants preserve the same abstract reasoning steps but change the words and context, any similarities that persist across carriers must come from the underlying logic, whereas the differences arise from surface semantics.

**Definition 5.1** (Natural Deduction System (Troelstra & Schwichtenberg, 2000; Pelletier & Hazen, 2024)). *A natural deduction system is a pair* $\mathsf{ND} = (F, R)$ *where:*

- *F: a formal language of formulas (e.g., propositional or first-order logic),*

- *R: a finite set of inference rules with introduction and elimination rules for each logical constant.*

*A derivation (or proof) in ND is a tree whose nodes are judgements of the form "a formula is derivable" and whose edges follow inference rules from R. Temporary assumptions may be introduced in sub-*

Table 1: Comparison of reasoning-flow similarities across LLMs. We report mean cosine similarity (position, velocity) and Pearson correlation (curvature) under 3 grouping criteria: logic, topic, and language. Results show that position similarity is dominated by surface carriers, while velocity and curvature highlight logical structure as the primary invariant. We also perform a random shuffle baseline on the order of logical inferences using Qwen3 0.6B. See Section 6 for more details.

| Model | Position Similarity | | | Velocity Similarity | | | Curvature Similarity | | |
|---|---|---|---|---|---|---|---|---|---|
| | Logic | Topic | Lang. | Logic | Topic | Lang. | Logic | Topic | Lang. |
| Qwen1.5 0.5B | 0.26 | 0.30 | **0.80** | **0.17** | 0.07 | 0.08 | **0.52** | 0.11 | 0.15 |
| Qwen2 0.5B | 0.21 | 0.24 | **0.85** | **0.15** | 0.06 | 0.07 | **0.52** | 0.10 | 0.12 |
| Qwen3 0.6B | 0.26 | 0.30 | **0.85** | **0.17** | 0.07 | 0.08 | **0.53** | 0.11 | 0.13 |
| Qwen3 1.7B | 0.44 | 0.46 | **0.89** | **0.19** | 0.08 | 0.09 | **0.46** | 0.13 | 0.15 |
| Qwen3 4B | 0.33 | 0.35 | **0.86** | **0.16** | 0.07 | 0.08 | **0.53** | 0.11 | 0.13 |
| LLaMA3 8B | 0.31 | 0.34 | **0.74** | **0.15** | 0.06 | 0.07 | **0.58** | 0.13 | 0.17 |
| Random Shuffle | 0.30 | 0.33 | **0.81** | 0.02 | 0.02 | **0.04** | 0.02 | 0.03 | **0.04** |

*derivations and are discharged by certain rules (e.g., $\rightarrow I$, $\neg I$). Each connective is governed by paired introduction and elimination rules, which together determine its proof-theoretic meaning.*

## 5.2 DATA DESIGN

To test whether LLM reasoning trajectories are governed by logical structure rather than semantic content, we generates parallel reasoning tasks that maintain identical logical scaffolding while systematically varying superficial characteristics, specifically topical domain and linguistic realization.

Our dataset construction employs a principled two-stage generation pipeline using GPT-5 (OpenAI, 2025). It proceeds as follows: (i) abstract logical templates are first constructed, followed by (ii) domain-specific and language-specific rewriting. Our final dataset comprises 30 distinct logical structures, each containing between 8 and 16 reasoning steps. Each logical structure is instantiated across 20 topical domains and realized in four languages (English, Chinese, German, and Japanese), yielding a total corpus of 2,430 reasoning sequences. This controlled design enables direct comparison of trajectories across logical forms and surface carriers, isolating the role of logical structure in embedding dynamics. Full generation prompts and sampled data cases are provided in Appendix E.

## 6 PLAY WITH LLMS

### 6.1 EXPERIMENTAL SETUP

We employ the Qwen3 (Yang et al., 2025) family models and LLaMA3 (Grattafiori et al., 2024). From the final transformer layer (before the LM head), we extract context-dependent hidden states $\{h_i^{(L)}\}$, where $h_i^{(L)} \in \mathbb{R}^d$ denotes the representation at layer $L$ and position $i$. Each reasoning step $x_t$ is a set of tokens indexed by $\mathcal{S}_t$, and its step-level embedding is defined by mean pooling: $y_t = \frac{1}{|\mathcal{S}_t|} \sum_{i \in \mathcal{S}_t} h_i^{(L)}, \quad y_t \in \mathbb{R}^d$. The resulting sequence $Y = (y_1, \ldots, y_T)$ forms the reasoning trajectory in representation space.

### 6.2 RESULTS ANALYSIS

**Similarity Table.** We evaluate LLMs by extracting hidden states across our dataset (Section 5) and computing similarities under three criteria: (i) **Logic**, grouping by deduction skeleton and averaging across topics and languages; (ii) **Topic**; and (iii) **Language**, both capturing surface carriers. This yields position, velocity, and curvature similarities (Table 1). Results show that logical similarity is low at zeroth order (position) but becomes dominant at first and second order (velocity and curvature), validating our hypothesis. Topic and language exhibit low velocity similarity, suggesting they might occupy orthogonal subspaces; by contrast, the high logical similarity at first and second order breaks this orthogonality, indicating that logical structure transcends surface carriers.

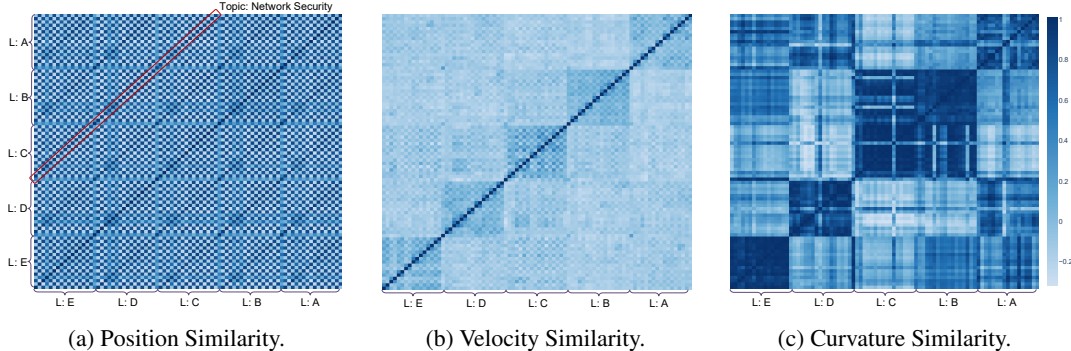

(a) Position Similarity.     (b) Velocity Similarity.     (c) Curvature Similarity.

Figure 2: Similarity of reasoning flows on Qwen3 0.6B. Blocks correspond to logic templates (L:A–E) instantiated with different topics and languages. (a) Position similarity (mean cosine): diagonals correspond to topics (e.g., Network Security), showing that positions are dominated by surface semantics. (b) Velocity similarity (mean cosine): semantic effects diminish, and flows with the same logical skeleton align while differing logics diverge. (c) Curvature similarity (Pearson): separation is further amplified, with logic emerging as the principal invariant and revealing close similarity between logics B and C. See Section 6 for more details.

**Random Shuffle.** We include a random shuffle baseline with Qwen3 0.6B in Table 1, where the order of logical steps (i.e., sentences) is randomly permuted and then fed into LLMs. The baseline performs poorly on velocity and curvature, indicating that the sequence order of reasoning is crucial for reasoning flow. In contrast, position similarity remains high, confirming that topic and language effects dominate surface semantics regardless of order. This contrast reinforces our view that higher-order geometric quantities, not raw embeddings, capture the invariant logical structure.

**Stable Scaling Effect.** Moreover, Table 1 shows two scaling axes: (1) model size (Qwen3 0.6B → 1.7B → 4B) and (2) model family (Qwen 1.5/2/3 and LLaMA3, varying in data and architecture). The similarity patterns remain remarkably stable. Increasing scale or switching families does not materially change the similarity measures. This consistency suggests the presence of a more general property of how LLMs internalize logical structure, independent of size or training recipe (Huh et al., 2024). We view this as a fascinating direction and plan to explore it further.

**Similarity Heatmap.** For visualization, we also analyze Qwen3 0.6B on a subset of our dataset (Figure 2). At the position level, embeddings cluster by topic and language. First-order differences reveal logical control: flows sharing the same skeleton align, while differing logics diverge even with identical carriers. Second-order curvature further amplifies this separation, and its strong cross-carrier consistency directly supports Proposition 4.11, confirming that logic governs reasoning velocity. Additional experiments across broader model families are presented in Appendix B.

**Challenging the Stochastic Parrots Argument.** Together, these results show that LLMs internalize latent logical structure beyond surface form, challenging the stochastic parrots (Bender et al., 2021) perspective. Specifically, the stochastic parrot argument suggests that LLMs do not perform genuine natural language understanding and cannot acquire meaning solely through next-token prediction (Bender et al., 2021; Bender & Koller, 2020). However, we provide evidence that learning from next-token prediction, augmented by standard post-training such as instruction tuning, can be sufficient to elicit semantic structure and support genuine natural language understanding. Moreover, we show that logical structure manifests in higher-order geometry of the representation space, which is difficult to explain under a purely surface-form view.

**Significance.** These findings suggest the existence of deeper, possibly universal constraints linking linguistic form to meaning. The generality, universality, and model-agnostic nature of the observed patterns align with the Platonic Representation Hypothesis (Huh et al., 2024; Jha et al., 2025), which posits that neural networks trained on different data and objectives nonetheless converge to shared underlying meanings in their representation spaces. In this view, LLMs acquire logical structure through next-token prediction because the most effective way to compress knowledge is to internalize and represent its underlying logic (Sutskever, 2023; Goldblum et al., 2024).

## 7 DISCUSSION

**Understanding, Not Generation.** Our theoretical framework focuses exclusively on natural language understanding. In NLP, generation presupposes understanding, so we restrict ourselves to the component that is conceptually cleaner and more amenable to theoretical analysis. Our goal is to articulate a post-hoc, model-agnostic, and training-algorithm-agnostic law describing how LLMs reason. Explaining why or how these reasoning patterns arise during training or generation is substantially more challenging and falls outside the scope of this work. Because we do not study generation, we also cannot meaningfully relate our geometric measures to generation-specific properties such as task output accuracy. Finally, uncovering the origins of these patterns would require analyzing the learning process itself—for example, learnability, training dynamics, detailed case studies, and more sophisticated experimental and dataset designs. These important questions are beyond the present scope and are left for future exploration.

**Contrast with Graph Perspective.** Prior works have modeled chain-of-thought reasoning as a graph structure (Minegishi et al., 2025; Wang et al., 2024a). While this provides a useful perspective, its predictive power is limited: graphs naturally suggest random walks between discrete nodes, which fits the noisy behavior of isolated embeddings but fails to capture the smooth, directed dynamics we observe under cumulative context. Our results in Section 6 show that well-trained LLMs learn flows governed by logical structure, transcending the surface semantics of language. Such continuity and logic-driven trajectories cannot be explained within a purely graph-based framework, but arise naturally in our differential-geometric view.

**Other Components in Learned Representation.** Beyond logical structure, learned representations also encode a wide spectrum of factors such as semantic objects, discourse tone, natural language identity, and even signals of higher-level cognitive behavior. Extending our framework to systematically isolate these components and characterize their interactions presents a major challenge for future work. A promising direction is to develop methods that disentangle additional attributes, enabling finer-grained insights into how language components co-evolve in representation space.

**Practical Implications.** Our results suggest that reasoning in LLMs unfolds as continuous flows, opening multiple directions. First, trajectory-level control offers principled tools for *steering, alignment, and safety*, extending vector-based interventions to flow dynamics (Venhoff et al., 2025; Chen et al., 2025; Gong et al., 2025; Huang et al., 2025; Bereska & Gavves, 2024). Second, our geometric view provides a formal framework to study abstract language concepts, enabling first-principle analyses of reasoning efficiency, stability, and failure modes. Third, it motivates new approaches to *retrieval and representation*, where embeddings respect reasoning flows rather than mere similarity, potentially improving RAG, reranking, and search (Weller et al., 2025; Cao et al., 2025). Finally, it hints at *architectural advances*, as models parameterizing latent flows may enable more efficient reasoning (Hao et al., 2024; Geiping et al., 2025; Zhang et al., 2025b; Shen et al., 2025).

## 8 CONCLUSION

We introduced a novel geometric framework that models LLM reasoning as smooth flows in representation space, with logic acting as a controller of local velocities. By disentangling logical structure from semantic carriers through a controlled dataset, we showed that velocity and curvature invariants reveal logic as the principal organizing factor of reasoning trajectories, beyond surface form. Our theory and experiments provide both a conceptual foundation and practical tools for analyzing reasoning, opening new avenues for interpretability.

## ACKNOWLEDGMENTS

ARZ was partially supported by NSF Grant CAREER-2203741 and NIH Grant R01HL169347. This research is supported by Google.org, the Google Cloud Research Credits program for the Gemini Academic Program, and Amazon AGI Labs SF.

LLMs USAGE STATEMENT

We clarify that LLMs were used solely as auxiliary aids, restricted to refining the manuscript's exposition for clarity and conciseness.

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

## CONTENTS

# Appendix

## A ADDITIONAL RELATED WORK

**Mechanistic Interpretability.** LLMs have exhibited unprecedented intelligence ever since their debut (OpenAI, 2022). Yet the underlying mechanisms remain opaque, as transformers are neural networks not readily interpretable by humans—motivating efforts to uncover why such capabilities emerge (Singh et al., 2024; Madsen et al., 2024). Mechanistic Interpretability (MI) pursues this goal by reverse-engineering transformer internals into circuits, features, and algorithms (Rai et al., 2024; Ferrando et al., 2024; Bereska & Gavves, 2024). The Transformer Circuits program at Anthropic exemplifies this agenda, systematically cataloging reusable computational subroutines (Anthropic, 2021). Empirical studies reveal concrete algorithmic mechanisms: grokking progresses along Fourier-like structures (Nanda et al., 2023), training can yield divergent solutions for the same task (Clock vs. Pizza) (Zhong et al., 2023), arithmetic emerges via trigonometric embeddings on helical manifolds (Kantamneni & Tegmark, 2025), and spatiotemporal structure is encoded through identifiable neurons (Gurnee & Tegmark, 2024). Beyond circuits, in-context learning and fine-tuning yield distinct representational geometries despite comparable performance (Doimo et al., 2024), while safety studies reveal polysemantic vulnerabilities where small-model interventions transfer to larger LLMs (Gong et al., 2025).

**Formal Logic with LLMs.** Recent work links transformer computation directly to logic. Log-precision transformers are expressible in first-order logic with majority quantifiers, providing an upper bound on expressivity (Merrill & Sabharwal, 2023), while temporal counting logic compiles into softmax-attention architectures, giving a constructive lower bound (Yang & Chiang, 2024). Beyond these characterizations, pre-pretraining on formal languages with hierarchical structure (e.g., Dyck) imparts syntactic inductive biases and improves efficiency (Hu et al., 2025). Synthetic logic corpora and proof-generation frameworks further strengthen reasoning, though benefits diminish as proofs lengthen (Morishita et al., 2024; Xia et al., 2025). Systematic evaluations, including LogicBench and surveys, highlight persistent failures on negation and inductive reasoning, despite partial gains from "thinking" models and rejection finetuning (Parmar et al., 2024; Jiang et al., 2025; Liu et al., 2025). In contrast, our work employs formal logic not as an end task, but as a tool to validate our geometric framework in LLMs' representation space, distinguishing our contribution from prior lines of work.

**Comparison with Park et al. (2025).** While Park et al. (2025) and our work both adopt a geometric lens for semantic and theoretical interpretability, we address fundamentally different phenomena using distinct mathematical frameworks:

- **Objective (Static Semantics vs. Dynamic Reasoning):** The primary motivation of Park et al. (2025) is to map the *static* organization of semantic knowledge. They investigate how fixed concepts (e.g., hierarchies like "animal" → "dog") are encoded relative to one another in the representation space. In contrast, our work investigates the *dynamic* process of reasoning. We model how internal representations evolve step-by-step as a model solves a logical problem, framing reasoning not as a static location but as a flow over time.

- **Geometric Nature (Structure vs. Motion):** Consequently, the specific geometries differ. Park et al. (2025) describe a structural geometry where hierarchical relations are encoded via *orthogonal subspaces* and categorical concepts appear as *polytopes* (simplices). Our work describes a kinematic geometry defined by *smooth flows*, analyzing properties of motion such as *velocity* and *curvature* along a reasoning trajectory.

- **Technique & Methodology:** Methodologically, Park et al. (2025) rely on linear algebraic tools, such as Linear Discriminant Analysis (LDA) and Causal Inner Products, to analyze static word unembeddings. Conversely, we develop a differential-geometric framework to analyze *context-cumulative* trajectories. By generating controlled datasets that isolate logical skeletons from semantic carriers, we show that logic acts as a *differential constraint* on the flow's velocity, rather than determining a static position in the subspace.

**Comparison with Li et al. (2025a).** While Li et al. (2025a) and our work both analyze the geometry of LLM representations, we diverge in our fundamental objects of study (understanding vs. generation) and in analytical scope (post-hoc laws vs. training dynamics):

- **Scope (Post-hoc Law of Reason vs. Training Dynamics):** Our goal is to articulate a *post-hoc, model-agnostic law* that describes how LLMs reason once fully trained. We treat the model as a fixed dynamical system to derive differential-geometric constraints on its internal flows. In contrast, Li et al. (2025a) focus on the *origins* of these patterns. They analyze the learning process itself, tracing how geometric properties (such as effective rank) evolve through distinct training phases like "entropy-seeking" and "compression-seeking". While they ask how representations evolve during *training*, we ask what invariant rules govern their trajectory during *inference*.

- **Domain (Natural Language Understanding vs. Generation):** Our framework focuses exclusively on Natural Language Understanding. Conversely, Li et al. (2025a) explicitly link geometric measures to *generation-specific* properties, correlating spectral shifts with $n$-gram memorization, cross-entropy loss, and output accuracy (e.g., pass@k). Because we do not study generation, we do not relate our measures to task output metrics, whereas validating these correlations is central to their empirical contribution.

- **Methodology (Theoretical Constraints vs. Empirical Correlations):** Methodologically, Li et al. (2025a) rely on empirical correlations between spectral metrics (RankMe, $\alpha$-ReQ) and downstream performance metrics across checkpoints. Our work is disjoint from this experimental design; we employ differential geometry to formalize logic as a velocity constraint on the reasoning flow. We leave the questions of learnability and training dynamics, central to Li et al. (2025a), outside our scope to focus on the geometry of the reasoning process itself.

# B  ADDITIONAL EXPERIMENTS

**More Similarity Heatmap.**  We additionally evaluate LLaMA3 (Grattafiori et al., 2024) and more Qwen3 (Yang et al., 2025) models (1.7B, 4B) to test robustness under the same experimental settings as in Section 6. The results (Figures 3, 4 and 5) confirm that our findings generalize across model sizes and families.

$C^1$ **Continuity Test.**  In Hypothesis 4.6, we posit that the discrete trajectory admits an underlying $C^1$ interpolation. This assumption serves to make the geometric framework well-defined; it does *not* assert that the model's internal computation unfolds in continuous time. Consequently, a rigorous empirical verification of $C^1$-regularity is impossible: language inputs are discrete, and the sequence index cannot be made infinitesimally small, as differentiability would require. Instead, we provide finite-difference–based smoothness diagnostics to illustrate that the observed embedding trajectory behaves consistently with this assumption. For a context-cumulative trajectory $Y = [y_1, \ldots, y_T]$ (Definition 4.5), we form velocities $v_t = y_{t+1} - y_t$ and examine their magnitudes $\|v_t\|$. As shown in Figure 6, the velocity norm behave consistently across the same MATH500 problem with six different answers presented in Figure 1, exhibiting no abrupt jumps. This provides visual support for the plausibility of our smooth-flow assumption. A more precise and formal theoretical justification is provided in Appendix D.

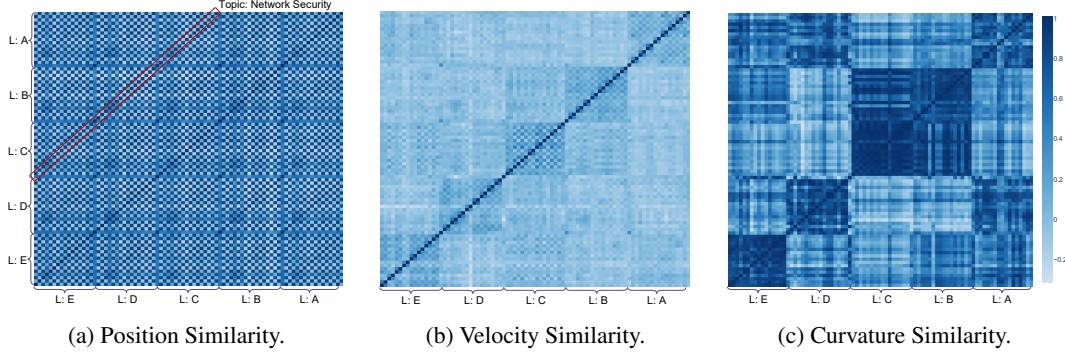

(a) Position Similarity.          (b) Velocity Similarity.          (c) Curvature Similarity.

Figure 3: Similarity of reasoning flows on Qwen3 1.7B.

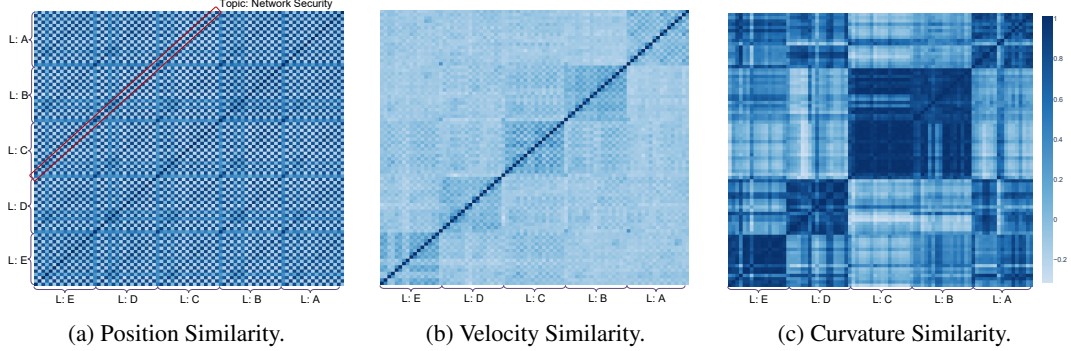

(a) Position Similarity.    (b) Velocity Similarity.    (c) Curvature Similarity.

Figure 4: Similarity of reasoning flows on Qwen3 4B.

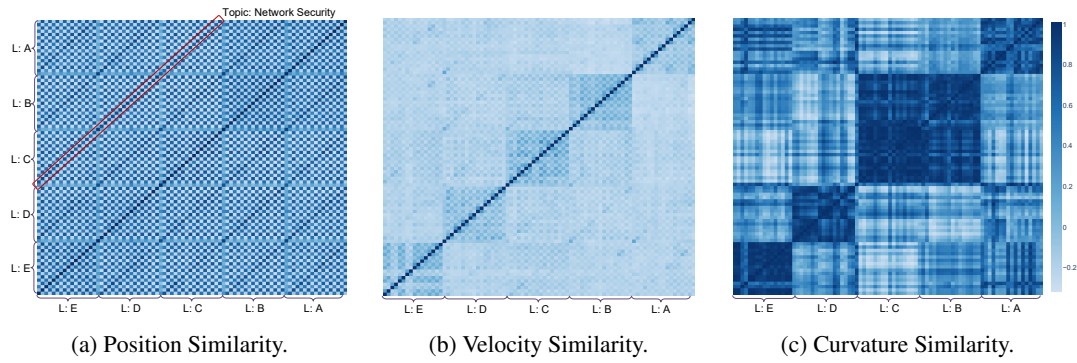

(a) Position Similarity.    (b) Velocity Similarity.    (c) Curvature Similarity.

Figure 5: Similarity of reasoning flows on Llama3 8B.

## C  SYMBOLIC GLOSSARY AND MAPPING RELATIONS

This section is a standalone roadmap that summarizes the spaces, maps, and commutative structure underlying our geometric view of reasoning.

### C.1  SPACES

- **Input space** $\mathcal{X}$ (often specialized to a vocabulary $\mathcal{V}$): discrete tokens/sentences.

- **Concept space** $\mathcal{C}$: abstract semantic space. A sentence $X$ is represented by a smooth *semantic trajectory*

$$\gamma_X : [0,1] \to \mathcal{M} \subseteq \mathcal{C},$$

  where $\mathcal{M}$ is a semantic submanifold for a coherent domain of meaning.

- **Representation space** $\mathcal{R} \subset \mathbb{R}^d$: the model's embedding space. Each prefix $X_t$ yields

$$y_t = \Psi(X_t) \in \mathbb{R}^d,$$

  sampling a continuous *representation trajectory* $\widetilde{\Psi} : [0,1] \to \mathbb{R}^d$.

- **Formal logical space** $\mathcal{L}_{\mathrm{form}}$: symbolic/human logic governed by a natural deduction system $\mathrm{ND} = (F, R)$, with formulas $F$ and rules $R$. Judgements and rule-based derivations live here.

- **Representation-based logical space** $\mathcal{L}_{\mathrm{rep}}$: the space of *reasoning increments in the embedding space*, defined by local variations of the trajectory, $\Delta y_t := y_{t+1} - y_t$. Geometric descriptors such as the Menger curvature $\kappa_t$ are evaluated here. This space is non-symbolic, and serves as the model's internal analogue of logic.

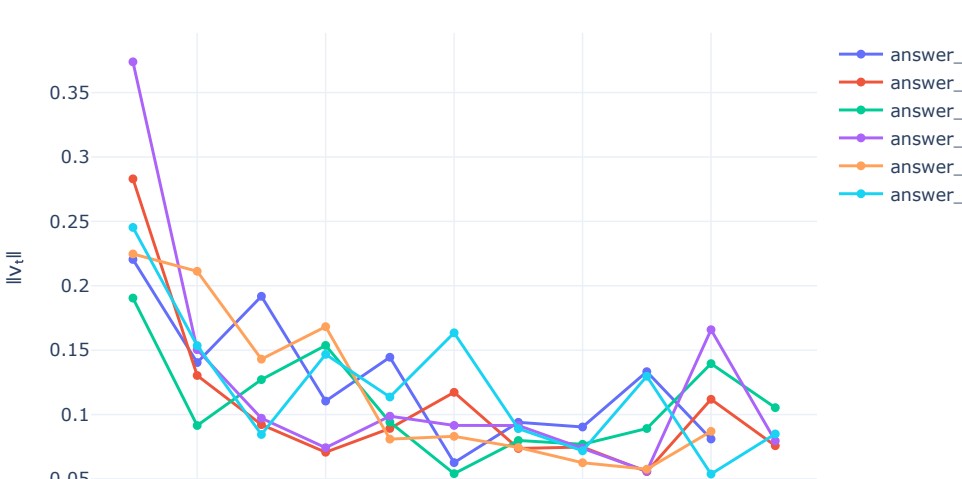

Figure 6: Velocity Norm $\|v_t\|$ for a MATH500 problem and its six answers.

## C.2 PRIMARY MAPS

- *Semantic interpretation*:

$$\Gamma : \ \mathcal{X} \to \mathrm{Curves}(\mathcal{C}), \qquad X \mapsto \gamma_X.$$

- *Neural representation*:

$$\Psi : \ \mathcal{X} \to \mathrm{Curves}(\mathcal{R}),$$

realized by token embeddings $\mathcal{E}$ and a contextual encoder $\Phi$, producing the continuous trajectory $\widetilde{\Psi}$ and sampled states $Y = (y_1, \ldots, y_T)$.

- *Canonical Alignment.*

**Definition C.1** (Canonical alignment map). *Assume $\Gamma$ and $\Psi$ are injective on the domain of interest. Define*

$$A := \Psi \circ \Gamma^{-1} : \ \mathrm{Curves}(\mathcal{C}) \to \mathrm{Curves}(\mathcal{R}).$$

*Then $A$ is a bijection between semantic curves and representation trajectories, and the top-level diagram commutes exactly:*

$$A \circ \Gamma = \Psi.$$

- *Flow vs. differential to logic.* We distinguish a human *flow operator* on concepts from a differential operator on representations:

$$F_{\mathcal{C}} : \ \gamma \mapsto (\text{human reasoning flow in } \mathcal{C}) \ \in \ \mathcal{L}_{\mathrm{form}}, \qquad D_{\mathcal{R}} : \ \widetilde{\Psi} \mapsto (\Delta y_t) \ \in \ \mathcal{L}_{\mathrm{rep}}.$$

The left operator $F_{\mathcal{C}}$ is *not* a discrete difference; it encodes how a semantic trajectory induces formal reasoning steps under ND. The right operator $D_{\mathcal{R}}$ extracts local increments from the representation trajectory.

## C.3 REASONING INCREMENTS AND CURVATURE

- **Formal side (concepts).** Human reasoning flow is captured at the semantic level by $F_{\mathcal{C}}$, which maps a semantic curve $\gamma$ into a sequence of formally valid steps in $\mathcal{L}_{\text{form}}$ per the rules ND.

- **Representation side (vectors).** The local increment $\Delta y_t = y_{t+1} - y_t$ encodes a step of representation flow in $\mathcal{L}_{\text{rep}}$.

- **Curvature as geometric intensity.** For three consecutive states $(y_{t-1}, y_t, y_{t+1})$, the Menger curvature

$$\kappa_t \;=\; c_M(y_{t-1}, y_t, y_{t+1}) \;=\; \frac{2\sqrt{1 - \text{CosSim}(u,v)^2}}{\|y_{t+1} - y_{t-1}\|}, \quad u := y_t - y_{t-1}, \; v := y_{t+1} - y_t,$$

couples angular change with scale, providing a geometry-aware proxy for the "strength" of a reasoning step in the representation.

## C.4 ROADMAP DIAGRAM

The overall structure can be read from the commutative roadmap below. Here $\mathcal{X}$ sits at the center; semantic and representation curves live to the left and right; formal and representation-based logics sit below. The top arrow is *strict* by definition of $A$; the vertical arrows express how each curve induces its respective notion of reasoning.

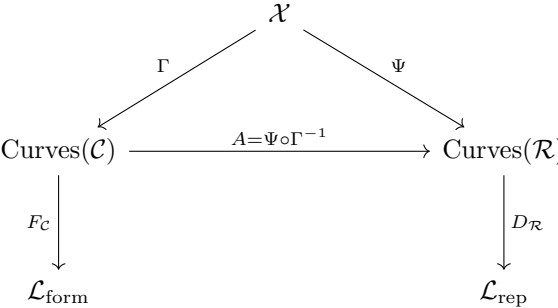

**Reading guide.** (1) Input sequences branch into a *semantic* curve (left) and a *representation* curve (right). (2) The canonical alignment $A = \Psi \circ \Gamma^{-1}$ identifies the two curves one-to-one. (3) The semantic curve induces human, rule-constrained steps in $\mathcal{L}_{\text{form}}$ via $F_{\mathcal{C}}$, while the representation curve induces vector increments in $\mathcal{L}_{\text{rep}}$ via $D_{\mathcal{R}}$. (4) Curvature in $\mathcal{L}_{\text{rep}}$ quantifies the geometric intensity of reasoning transitions and can be related back to formal steps under appropriate correspondences established elsewhere in the paper.

# D GEOMETRIC FOUNDATIONS OF REASONING TRAJECTORIES

In this section, we establish the geometric foundations for analyzing reasoning as smooth flows in representation space. We first construct representation trajectories as $C^1$ curves via a relaxed prefix-mask mechanism, thereby justifying smoothness as a working principle. Then, we introduce Menger curvature as a computable descriptor that couples angular deviation with distance variation, providing a principled measure of the intensity of reasoning turns.

## D.1 CONTINUITY OF REPRESENTATION TRAJECTORIES

In this section, we provide a rigorous and explicit construction of a $C^1$ trajectory using a relaxed prefix-mask mechanism. This construction justifies our working assumption that representation trajectories are $C^1$. Note that the symbol $\mathcal{I}$ (Definition 3.2) is defined with a slight variation compared to main paper: here it is specialized to encode positional information, while the remaining complexities of the model architecture are subsumed into a single mapping $\Phi$.

**Definition D.1** (Neural Encoding View of Sentence Representation). *Let $x = (u_1, \ldots, u_n)$ be a sentence with tokens $u_i$ drawn from a vocabulary space $\mathcal{V}$. Define an embedding map*

$$\mathcal{E} : \mathcal{V} \to \mathbb{R}^d, \qquad u_i \mapsto \mathcal{E}(u_i),$$

*which assigns each token a $d$-dimensional vector. Augmenting $\mathcal{E}(u_i)$ with positional information yields the input sequence*

$$z_0 = \big(\mathcal{E}(u_1), \mathcal{E}(u_2), \ldots, \mathcal{E}(u_n)\big) \in (\mathbb{R}^d)^n.$$

*Let $\Phi : (\mathbb{R}^d)^n \times \mathcal{I} \to \mathbb{R}^d$ denote a contextual encoder that maps a sequence of token embeddings together with positional information to a global sentence-level representation, where $\mathcal{I}$ is the positional encoding space and $\mathcal{I}_n \subset \mathcal{I}$ denotes the set of encodings for the first $n$ positions. For a fixed $\iota = (\iota_1, \ldots, \iota_n) \in \mathcal{I}_n$, we define*

$$\Psi(x) := \Phi\big(z_0, \iota_n\big) = \Phi\big(\mathcal{E}(u_1), \ldots, \mathcal{E}(u_n), \iota\big) \in \mathbb{R}^d.$$

*In this view, $\Psi$ subsumes both the static token embeddings and the contextual transformations carried out by the neural network.*

*Hence the hidden state $y_t = \Psi(S_t)$ in Definition 4.5 should be interpreted not merely as a sum of embeddings, but as the outcome of the full encoding process applied to the prefix $S_t$.*

*Mask-aware realization (for later use). Fix a maximum length $N \geq n$ and consider the mask-aware realization of the same encoder,*

$$\Phi_m : (\mathbb{R}^d)^N \times \mathcal{I}_N \times \{0, 1\}^N \to \mathbb{R}^d,$$

*such that for any length $n \leq N$,*

$$\Phi_m\big((\mathcal{E}(u_1), \ldots, \mathcal{E}(u_n), 0, \ldots, 0, \iota), \mathbb{1}_{\{i \leq n\}}\big) := \Phi\big(\mathcal{E}(u_1), \ldots, \mathcal{E}(u_n), (\mathbb{1}_{\{i \leq n\}} \iota_i)_{i=1}^N\big).$$

*When the mask is all ones on $\{1, \ldots, n\}$, this coincides with the above definition; when we pass a mask explicitly we will write $\Phi(\cdot, M)$.*

**Hypothesis D.2** (Smooth Trajectory Hypothesis). *The sequence of representations $y_t = \Psi(X_t)$ generated during a reasoning process lies on a smooth, differentiable trajectory in the embedding space.*

**Definition D.3** (Relaxed-Mask Sentence Representation). *Let each sentence in Hypothesis 4.4 be $x_t = (u_{t,1}, \ldots, u_{t,n_t})$ for $t = 1, \ldots, T$, and let the full token stream be*

$$U_{1:N} = (u_{1,1}, \ldots, u_{1,n_1}, u_{2,1}, \ldots, u_{2,n_2}, \ldots, u_{T,1}, \ldots, u_{T,n_T}),$$

*with total length $N = \sum_{t=1}^T n_t$ and cumulative lengths $N_t = \sum_{j=1}^t n_j$. Introduce a continuous progress parameter $s \in [0, 1]$ and a relaxed prefix mask*

$$m_s : \{1, \ldots, N\} \to [0, 1],$$

*which specifies the fractional inclusion of each token at progress $s$.*

*Using the embedding map $\mathcal{E}$ and positional information $\mathcal{I}_N$ from Definition D.1, define the masked input sequence at progress $s$ by*

$$z_s = \big(m_s(i)\, \mathcal{E}(u_i)\big)_{i=1}^N, \qquad \iota^s = \big(m_s(i)\, \iota_i\big)_{i=1}^N.$$

*and the associated hard mask*

$$M_s(i) := \mathbb{1}_{\{m_s(i)=1\}}, \qquad i = 1, \ldots, N.$$

*Let $k(s) := \lceil sN \rceil$, denote the number of tokens included at progress $s$. The truncated masked sequences are then defined as*

$$z_s^{(\leq k)} := (z_s(1), \ldots, z_s(k(s))) \in (\mathbb{R}^d)^{k(s)}, \qquad \iota^{s,(\leq k)} := (\iota^s(1), \ldots, \iota^s(k(s))) \in \mathcal{I}^{k(s)}.$$

With the mask-aware encoder $\Phi_m : (\mathbb{R}^d)^N \times \mathcal{I}^N \times \{0,1\}^N \to \mathbb{R}^d$ introduced above, the continuous representation trajectory is defined by

$$\widetilde{\Psi}(s) := \Phi_m(z_s, \iota^s, M_s) \in \mathbb{R}^d, \quad where \ \Phi_m(z_s, \iota^s, M_s) := \Phi\big(z_s^{(\leq k)}, \iota^{s,(\leq k)}\big).$$

At sentence boundaries $s_t := N_t/N$, the hard prefix mask is recovered exactly by choosing a smooth function with flat tails (see Proposition D.4); consequently,

$$y_t = \Psi(S_t) = \Phi\big(z_{s_t}, \iota^{s_t}, M_{s_t}\big) = \widetilde{\Psi}(s_t), \qquad t = 1, \ldots, T.$$

**Proposition D.4** (Continuity of the Relaxed-Mask Trajectory). *Suppose the relaxed mask takes the form*

$$m_s(i) = g(sN - i),$$

*where $g \in C^\infty(\mathbb{R})$ satisfies $g(x) = 0$ for $x \leq -\delta$, $g(x) = 1$ for $x \geq \delta$, with some $0 < \delta < \frac{1}{2}$ (i.e., a smoothstep/bump with flat tails). Assume the encoder $\Phi$ is $C^1$. Then the mapping $\widetilde{\Psi} : [0,1] \to \mathbb{R}^d$ defines a $C^1$ trajectory in embedding space. Moreover, the discrete sentence embeddings $(y_t)_{t=1}^T$ are exactly samples of this trajectory at $s_t = N_t/N$:*

$$y_t = \widetilde{\Psi}(s_t), \qquad t = 1, \ldots, T.$$

*Proof.* For each token $U_i$, we define the masked embedding and positional encoding as

$$(z_s(i), \iota^s(i)) \;=\; m_s(i)\big(\mathcal{E}(U_i), \iota_i\big) = g(sN - i)\big(\mathcal{E}(U_i), \iota_i\big).$$

Since $g$ is $C^\infty$ and both $\mathcal{E}(U_i)$ and $\iota_i$ are constant in $s$, each coordinate pair $(z_s(i), \iota^s(i))$ varies smoothly with $s$. Hence the entire masked sequence

$$(z_s, \iota^s) \;=\; \big(z_s(1), \ldots, z_s(N); \ \iota^s(1), \ldots, \iota^s(N)\big)$$

is a smooth trajectory with respect to $s$. The mask $M_s(i) = \mathbb{1}_{\{m_s(i)=1\}}$ is piecewise constant in $s$ and equals the all-ones indicator on indices where $sN - i \geq \delta$, and zeros where $sN - i \leq -\delta$; in particular, it is locally constant on neighborhoods that avoid the transition band $|sN - i| < \delta$.

By assumption, $\Phi$ is a composition of affine maps, matrix multiplications, LayerNorm, residual connections, softmax attention, and smooth pointwise nonlinearities. As a function of its inputs, such a network is smooth; thus, on any interval where $M_s$ is fixed, the composite map

$$\widetilde{\Psi}(s) = \Phi\big(z_s, \iota^s, M_s\big)$$

is $C^1$ by the chain rule.

At sentence boundaries $s_t = N_t/N$, choose $\delta < \frac{1}{2}$ so that $g(N_t - i) = 1$ for $i \leq N_t$ and $g(N_t - i) = 0$ for $i \geq N_t + 1$. Hence $m_{s_t}(i) \in \{0,1\}$ exactly and $M_{s_t}(i) = \mathbb{1}_{\{i \leq N_t\}}$. Substituting into the definition,

$$\widetilde{\Psi}(s_t) = \Phi\big((\mathcal{E}(U_1), \ldots, \mathcal{E}(U_{N_t}), 0, \ldots, 0, \iota), \mathbb{1}_{\{i \leq N_t\}}\big) = \Psi(S_t) = y_t,$$

which shows that the discrete embeddings $(y_t)_{t=1}^T$ are precisely samples of the continuous trajectory $\widetilde{\Psi}(s)$. $\square$

**Remark D.5.** *Since $\Phi(\cdot)$ implemented with affine maps, matrix multiplications, LayerNorm, residual connections, softmax attention, and smooth pointwise nonlinearities (e.g., GELU/SiLU/Swish), it's reasonable to assume that is $C^1$. If ReLU activations (or other piecewise smooth nonlinearities) are used instead of smooth ones, the mapping $\widetilde{\Psi}$ remains continuous and is differentiable almost everywhere. Since this does not affect the manifold-level geometric reasoning, we idealize $\Phi$ as smooth throughout our discussion.*

*The construction above is merely one possible realization of a continuous and $C^1$ trajectory $\widetilde{\Psi}(s)$. In fact, many alternative constructions are possible. This abundance of realizations justifies our assumption that the sentence $\Psi(X_T)$, through its step-by-step variations, can be viewed as $T$ points lying on a smooth, differentiable curve. On this basis, we can consistently define the notion of flow velocity in Definition 4.10.*

### D.2   MENGER CURVATURE

**Definition D.6** (Menger Curvature). *Let $x_1, x_2, x_3 \in \mathbb{R}^n$ be three distinct points. The* Menger curvature *of the triple $(x_1, x_2, x_3)$ is defined as the reciprocal of the radius $R(x_1, x_2, x_3)$ of the unique circle passing through the three points:*

$$c(x_1, x_2, x_3) \;=\; \frac{1}{R(x_1, x_2, x_3)}.$$

**Proposition D.7** (Computation Formula). *Let $a = \|x_2 - x_3\|$, $b = \|x_1 - x_3\|$, and $c = \|x_1 - x_2\|$. Denote by $\Delta(x_1, x_2, x_3)$ the area of the triangle spanned by the three points. Then the circumradius $R$ and the Menger curvature $c(x_1, x_2, x_3)$ are given by*

$$R(x_1, x_2, x_3) \;=\; \frac{abc}{4\Delta(x_1, x_2, x_3)}, \qquad c(x_1, x_2, x_3) \;=\; \frac{4\Delta(x_1, x_2, x_3)}{abc}.$$

*Proof.* The formula follows from classical Euclidean geometry: for a triangle with side lengths $a, b, c$ and area $\Delta$, the circumradius satisfies $R = \frac{abc}{4\Delta}$. Taking the reciprocal yields the Menger curvature. $\qquad\square$

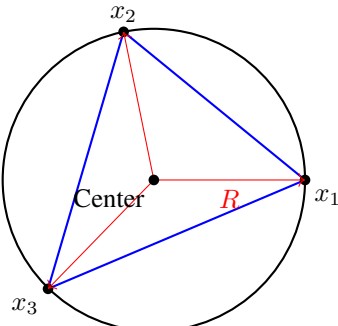

Figure 7: Circumcircle through three points $x_1, x_2, x_3$, with radius $R$ and Menger curvature $1/R$.

**Proposition D.8** (Menger curvature from three consecutive states). *Let $y_{t-1}, y_t, y_{t+1} \in \mathbb{R}^d$ be three distinct points and set*

$$u := y_t - y_{t-1}, \qquad v := y_{t+1} - y_t.$$

*Write the side lengths*

$$a = \|u\|, \quad b = \|v\|, \quad c = \|v - u\| = \|y_{t+1} - y_{t-1}\|.$$

*The Menger curvature of the triple $(y_{t-1}, y_t, y_{t+1})$ equals*

$$c_M(y_{t-1}, y_t, y_{t+1}) = \frac{4\,\Delta(y_{t-1}, y_t, y_{t+1})}{abc} = \frac{2\sqrt{1 - \mathrm{CosSim}(u, v)^2}}{\|y_{t+1} - y_{t-1}\|},$$

*where $\mathrm{CosSim}(u, v) := \dfrac{\langle u, v\rangle}{\|u\|\,\|v\|}$. (If the three points are collinear, $c_M := 0$.)*

*Proof.* By classical Euclidean geometry, for a triangle with side lengths $a, b, c$ and area $\Delta$, the circumradius satisfies $R = \dfrac{abc}{4\Delta}$. The Menger curvature is the reciprocal $c_M = 1/R = \dfrac{4\Delta}{abc}$.

It remains to express $\Delta$ in terms of $u$ and $v$. The (unsigned) area of the triangle spanned by $u$ and $v$ can be written in a dimension-independent way via the Gram determinant:

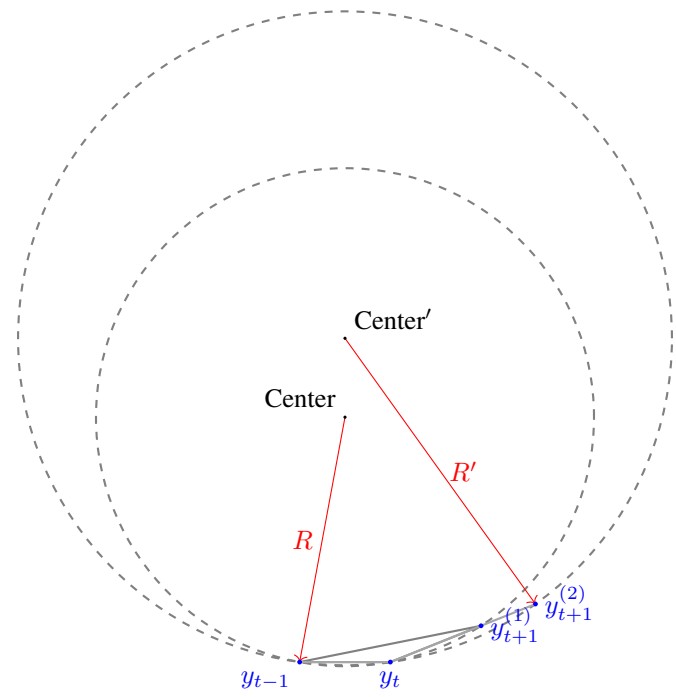

Figure 8: Two circumcircles through $\{y_{t-1}, y_t, y_{t+1}^{(1)}\}$ and $\{y_{t-1}, y_t, y_{t+1}^{(2)}\}$, with radii $R$ and $R'$. Here $y_{t+1}^{(1)}$ and $y_{t+1}^{(2)}$ lie on the same ray from $y_t$.

$$\Delta = \frac{1}{2}\|u \wedge v\| = \frac{1}{2}\sqrt{\det\begin{pmatrix} \langle u, u \rangle & \langle u, v \rangle \\ \langle v, u \rangle & \langle v, v \rangle \end{pmatrix}} = \frac{1}{2}\sqrt{\|u\|^2\|v\|^2 - \langle u, v \rangle^2}.$$

Substituting $a = \|u\|$, $b = \|v\|$, $c = \|v - u\|$ into $c_M = \dfrac{4\Delta}{abc}$ gives

$$c_M = \frac{2\sqrt{\|u\|^2\|v\|^2 - \langle u, v \rangle^2}}{\|u\|\,\|v\|\,\|v - u\|}.$$

Divide the numerator and denominator by $\|u\|\,\|v\|$ and denote $s := \mathrm{CosSim}(u, v) = \dfrac{\langle u, v \rangle}{\|u\|\,\|v\|}$. Then

$$c_M = \frac{2\sqrt{1 - s^2}}{\|v - u\|} = \frac{2\sin\theta}{c},$$

where $\theta$ is the angle between $u$ and $v$ (so $\sin\theta = \sqrt{1 - s^2}$). If the three points are collinear, $\Delta = 0$ and hence $c_M = 0$, consistent with the convention. This proves the claim. $\qquad\square$

**Remark D.9.** *As illustrated in Figure 8, using the Menger curvature instead of cosine similarity is significant. Cosine similarity only depends on the angle at $y_t$, so the two triples $\{y_{t-1}, y_t, y_{t+1}^{(1)}\}$ and $\{y_{t-1}, y_t, y_{t+1}^{(2)}\}$ would look identical. In contrast, their circumradii $R$ and $R'$ are different, hence the Menger curvatures distinguish two different curvature regimes. This demonstrates how Menger curvature captures both angle and length information, enabling discrimination that cosine similarity alone cannot provide.*

# E  DATA GENERATION

We provide the exact prompt templates and the representative sampled data instances used in our data generation process. The two-stage pipeline is run with GPT-5.

## E.1  PROMPTS FOR DATA GENERATION

The following prompts are used for abstract logical templates construction and domain-specific and language-specific rewriting.

---

**Prompt for Logic Pattern Generation**

You are a formal logic pattern generator.
Goal: Create an abstract, domain-agnostic reasoning sequence of exactly N steps, written in symbolic form, using standard propositional/first-order logic notation.
**Strict output format:**

- Exactly N lines, each line starts with a bracketed index and a single formula or conclusion, e.g.:

```
[1] A -> B
[2] B -> C
[3] C -> D
[4] (D & E) -> F
[5] forall x(H(x) -> J(x))
[6] A
[7] E
[8] H(a)
[9] D  (from [1-3] and [6])
[10] F & J(a)  (from [4],[7],[5],[8],[9])
```

- Use only symbols from: $\neg, \wedge, \vee, \rightarrow, \leftrightarrow, \forall, \exists$, parentheses, predicate letters with uppercase (A,B,C,...) and predicate symbols like H(x), J(x).

- You may include brief justifications at the end of lines in parentheses referencing earlier step indices (e.g., (from [2] and [5])).

- The sequence must be internally coherent (later steps can be derived from earlier ones), but no proof of a fixed target is required.

- No extra commentary before or after the lines. No natural-language sentences.

**Parameters (provided by caller):**

- N: number of steps to output.

- logic: a label for this abstract logic (optional).

N = {N}
logic = {logic}
Now produce exactly N lines.

---

---

**Prompt for Reasoning Rewriter**

You are a reasoning rewriter.

**Task:** Given an abstract N-step reasoning scaffold (formal symbolic lines) and a target topic, rewrite the scaffold into a topic-specific natural-language reasoning sequence with exactly the same number of steps and the same dependency structure.

**Inputs (provided by caller):**

- `Topic`: the target domain (e.g., weather, software).
- `Abstract Steps (1..N)`: the neutral scaffold, numbered 1..N.
- `N`: the total number of steps.

**Output requirements:**

- Produce exactly N steps, each line begins with the same bracketed index as the abstract: `[1] ...` to `[N] ....`.
- Keep step count and ordering identical to the abstract. Do not merge, split, add, or remove steps.
- Preserve the logical dependencies: if abstract step k enables k+1, your rewrite must preserve that relationship in the topic.
- Use concrete domain terms appropriate to the topic, but keep sentences concise and precise.
- No extra commentary before or after the steps.

**Multilingual mode (when `Languages:` are specified by the caller):**

- Create a separate section for each requested language code.
- Each section starts with a header line `===  ===` (e.g., `=== en ===`).
- Under each header, write the N steps with bracketed indices `[1] .. [N]` in that language.
- Keep the content aligned across languages (same meaning per step index).

**Inputs you will receive:**

Topic: {topic}
Abstract Steps (1..N): {ABSTRACT_STEPS}
N = {N}
Now perform the rewrite.

---

## E.2 DATA EXAMPLES

Table 2 presents a 9-step logical scaffold from our dataset. We illustrate its instantiation in two distinct domains, weather and finance, providing the corresponding statements in both English (EN) and German (DE).

Table 2: Logic Example (9-Step) with Weather and Finance Topics in English and German

| Abstract Logic | Topic: Weather | Topic: Finance |
|---|---|---|
| [1] $A \rightarrow B$ | **EN:** If moisture converges over the city, then thunderclouds develop. **DE:** Wenn über der Stadt Feuchte konvergiert, dann bilden sich Gewitterwolken. | **EN:** If the firm's interest coverage ratio exceeds 3.0x, then the firm is deemed able to meet interest obligations. **DE:** Wenn die Zinsdeckungskennzahl des Unternehmens über 3,0x liegt, dann gilt das Unternehmen als fähig, Zinszahlungen zu leisten. |
| [2] $B \rightarrow C$ | **EN:** If thunderclouds develop, then heavy rain occurs. **DE:** Wenn sich Gewitterwolken bilden, dann tritt starker Regen auf. | **EN:** If the firm is deemed able to meet interest obligations, then the bank will approve a new term loan. **DE:** Wenn das Unternehmen als fähig gilt, Zinszahlungen zu leisten, dann wird die Bank ein neues Laufzeitdarlehen genehmigen. |
| [3] $\forall x(H(x) \rightarrow J(x))$ | **EN:** For any location x, if a cold front passes x, then temperatures drop at x. **DE:** Für jeden Ort x gilt: Wenn eine Kaltfront x überquert, dann sinkt dort die Temperatur. | **EN:** For any security x, if x is a U.S. Treasury, then x is acceptable as repo collateral. **DE:** Für jedes Wertpapier x gilt: Wenn x eine US-Staatsanleihe ist, dann ist x als Repo-Sicherheit zulässig. |
| [4] $H(a)$ | **EN:** A cold front is passing the airport. **DE:** Eine Kaltfront überquert den Flughafen. | **EN:** Bond A is a U.S. Treasury. **DE:** Anleihe A ist eine US-Staatsanleihe. |
| [5] $A$ | **EN:** Moisture is converging over the city. **DE:** Über der Stadt herrscht Feuchtekonvergenz. | **EN:** The firm's interest coverage ratio exceeds 3.0x. **DE:** Die Zinsdeckungskennzahl des Unternehmens liegt über 3,0x. |
| [6] $B$ (from [1], [5]) | **EN:** From [1] and [5], thunderclouds develop. **DE:** Aus [1] und [5] folgt, dass sich Gewitterwolken bilden. | **EN:** The firm is deemed able to meet interest obligations (from [1] and [5]). **DE:** Daher gilt das Unternehmen als fähig, Zinszahlungen zu leisten (aus [1] und [5]). |
| [7] $C$ (from [2], [6]) | **EN:** From [2] and [6], heavy rain occurs. **DE:** Aus [2] und [6] folgt, dass starker Regen auftritt. | **EN:** The bank will approve a new term loan (from [2] and [6]). **DE:** Daher wird die Bank ein neues Laufzeitdarlehen genehmigen (aus [2] und [6]). |
| [8] $J(a)$ (from [3], [4]) | **EN:** From [3] and [4], temperatures drop at the airport. **DE:** Aus [3] und [4] folgt, dass am Flughafen die Temperatur sinkt. | **EN:** Bond A is acceptable as repo collateral (from [3] and [4]). **DE:** Daher ist Anleihe A als Repo-Sicherheit zulässig (aus [3] und [4]). |
| [9] $C \wedge J(a)$ (from [7], [8]) | **EN:** From [7] and [8], heavy rain occurs and temperatures drop at the airport. **DE:** Aus [7] und [8] folgt: Es tritt starker Regen auf und am Flughafen sinkt die Temperatur. | **EN:** The bank will approve a new term loan and Bond A is acceptable as repo collateral (from [7] and [8]). **DE:** Somit wird die Bank ein neues Laufzeitdarlehen genehmigen und Anleihe A ist als Repo-Sicherheit zulässig (aus [7] und [8]). |

