# OpenReview forum: "The Geometry of Reasoning: Flowing Logics in Representation Space"
_ICLR.cc/2026/Conference — ICLR 2026 Poster_

### Official Review · Reviewer_xzaB · 2025-10-15

**Soundness:** 2
**Presentation:** 1
**Contribution:** 3
**Rating:** 6
**Confidence:** 3

**Summary:**

The paper analyzes “reasoning” in LLMs, particularly the reasoning flows (trajectories) in representation space.
Logic is posited to act as a local controller of velocity along these flows, with Menger curvature used to capture second-order structure.
The authors construct a dataset that keeps formal logical skeletons fixed while varying topical and linguistic “carriers”, then extract hidden states from Qwen3/LLaMA3 to compare position, first-order/second-order differences and similarities. They report that while positions cluster by surface semantics, velocity/curvature similarities align by logic, supporting the claim that logic governs flow dynamics.

**Strengths:**

- An interesting work on reasoning, the authors view logic as a carrier-invariant framework for reasoning, and they test if LLMs have learned these structural invariants in their own embedding space.

- Empirical comparisons across model sizes support qualitative claims.

- Some interesting findings, for example, in the representation space, sentences on the same topic cluster together. However, when looking at the differences of curvature, logical structure emerges as the dominant factor even across unrelated topics and languages.

**Weaknesses:**

- The paper is not easy to follow. Many key notions are introduced before being properly defined or motivated, which makes it difficult for readers outside the narrow intersection of geometry and logic to follow the argument. For example, the term “flow” appears early (Abstract, intro) for many times, but its formalization as a sequence of hidden states, or its mapping to logical entailment steps, is only explained several pages later (Sec. 4.2). Similarly, “menger curvature” is used as a core analytic measure before the geometric intuition or connection to reasoning trajectories is established. Also, it is often mentioned without details: what the Menger curvature is, how it is calculated, or at least some intuition should be provided. A short schematic or example trajectory would help. Recent ICLR/ICML papers have handled similar conceptual density more clearly, see, for instance, “The Geometry of Categorical and Hierarchical Concepts in LLMs” (Park et al., ICLR 2025) and “Tracing the Representation Geometry of Language Models from Pretraining to Post-training” (Zhang et al., ICML 2025).

- The paper cites several strands but does not sufficiently contrast contributions with recent, closely related geometry/trajectory work at ICLR/ICML.

- Its evidential basis is correlational and methodologically fragile in its current form. For example, the smoothness hypothesis needs independent validation. Hyp. 4.6 is asserted with a construction in App. C.1, but the fitted smooth curve could be an artifact. Maybe consider reporting results on shuffled/phase-randomized controls as baseline.
Maybe also consider adding causal tests, stronger statistical treatment, external benchmark dataset validation, and clearer positioning/discussion vs. recent ICLR/ICML work; this submission could become a citable contribution to the geometry-of-reasoning literature. But I'm not sure if it can or will be done in the CR.

**Questions:**

- It is not clear that the dataset isolates formal logic from its semantic context. There is a lack of systematic analysis, e.g., statistical analysis, to demonstrate the separation. In other words, how to ensure the disentanglement of the logic format and the semantic context?

---

> ### Author Response · Authors · 2025-11-21
> **Rebuttal (Part 1)**
>
> Thank you for appreciating our work as interesting! Here we address your concerns specifically.
>
> ## W1: The paper is not easy to follow.
> Thank you for the comment and suggestion. Our writing style reflects the structure of many theoretical papers: the high-level intuition is introduced first, and the precise mathematical definitions follow. It might be hard to understand at first glance for readers come from different backgrounds.
>
> As pointed out by other reviewers (`p3p6, 2CaL, umD2`), our paper is **well written with a good presentation**. Reviewer `2CaL` also notes that, compared to the **extremely complicated and difficult problem of interpreting LLM hidden states in relation to human thinking**, our approach offers a clearer and more tractable perspective. You are right that some key concepts are introduced early and formally defined later. This is **intentional for readability**, and all variables and constructs are fully formalized in **Sections 3 and 4**.
>
> Regarding Menger curvature: we provide a precise and explicit definition in **Definition 3.4**, and additional computational details and intuition are given in **Appendix D.2**, together with schematic illustrations in **Figures 7 and 8**. We have added additional forward pointers in the introduction (**Line 053**) to help guide readers to these definitions earlier.
>
> We are happy to clarify if you have more questions. Feel free to ask during the rebuttal phase!
>
> ## W2: The paper cites several strands but does not sufficiently contrast contributions with recent, closely related geometry/trajectory work at ICLR/ICML.
>
> Thank you for the feedback. We have expanded the comparison with recent geometry-based interpretability work by adding a dedicated section in Appendix A (**Lines 867–917**). This new section provides a detailed, point-by-point contrast with two of the most relevant works.
>
> ### Comparison with Park et al. (2025) [1] (Lines 872–892)
>
> We highlight three major differences:
> - **Objective:** Park et al. study static semantic geometry, focusing on how hierarchical concepts (“animal $\to$ dog”) are embedded. Our work studies the dynamic geometry of reasoning, modeling how embeddings evolve during a logical inference process. This shift from static semantics to dynamic reasoning is fundamental.
> - **Geometric Nature:** Park et al. characterize orthogonal subspaces and polytopic structures in static embeddings. We characterize smooth flows and use velocity and curvature to capture the motion of representations. Thus, their geometry is structural; ours is kinematic/differential.
> - **Methodology:** Park et al. rely on Linear Discriminant Analysis and Causal Inner Products over static word unembeddings. We build a differential-geometric framework over context-cumulative trajectories and explicitly show how logic acts as a constraint on the flow field, not on static positions.
>
>
> ### Comparison with Li et al. (2025a) [2] (Lines 896–917)
> We clarify divergences in scope, domain, and methodology:
> - **Scope:** Li et al. analyze training dynamics, explaining how geometric properties emerge across training phases. We provide a post-hoc, model-agnostic law describing reasoning behavior of fully trained models, independent of how they were trained.
> - **Domain:** Li et al. study generation, correlating spectral metrics with memorization and task accuracy. We study Natural Language Understanding only, as stated in Section 7 Line 488-498. Our geometric measures are not tied to generation metrics like loss or pass@k.
> - **Methodology:** Li et al. rely on empirical correlations (e.g., RankMe, α-ReQ) across checkpoints. We use theoretical constraints on the flow (velocity, curvature) derived from differential geometry, isolating logical skeletons via controlled datasets. Our framework is therefore fundamentally different both in analytic technique and interpretive intent.

---

> > ### Author Response · Authors · 2025-11-21
> > **Rebuttal (Part 2)**
> >
> > ## W3: Its evidential basis is correlational and methodologically fragile in its current form.
> > We appreciate the reviewer’s suggestions. We have strengthened the empirical analysis along several dimensions:
> >
> > - **(1) Hypothesis 4.6 validation.**
> >
> > Hypothesis 4.6 is introduced solely to make the geometric framework **well defined**; it does not claim that the model’s internal computation unfolds in continuous time. Consequently, a rigorous empirical verification of $C^1$-regularity is **impossible**: language inputs are discrete, and the sequence index cannot be made infinitesimally small, as differentiability would require.
> >
> > However, we added additional diagnostics for the $C^1$ continuity hypothesis in **Lines 927–937 and Fig. 6**, including finite-difference smoothness tests across six independent answers to the same MATH500 problem. The velocity norms show no abrupt jumps, supporting the plausibility of the smooth-flow assumption.
> >
> > - **(2) Shuffled/phase-randomized baseline.**
> >
> >  Per your request, we added a random-shuffle baseline in the **updated Table 1** and discussed it in **Lines 453–458**. When we destroy the logical order, velocity and curvature collapse, while position similarity remains high. Thus, **the sequence order of reasoning is crucial for reasoning flow**. This sharp contrast shows that higher-order geometric quantities, rather than raw embeddings, track the invariant logical structure. This confirms our finding and theoretical framework.
> >
> > - **(3) Benchmarks and external datasets.**
> >
> >  External benchmarks are not suitable for validating our theory. Our goal is to provide post-hoc geometric evidence that logic acts as a differential constraint on embedding flows, and this requires isolating logical structure from surface semantics. **Public datasets do not control for logical skeletons or carrier variation, which is why we build our own controlled dataset.** As discussed in **Lines 488–498**, our framework focuses exclusively on **Natural Language Understanding** and aims to reveal the geometry of reasoning, not to compete on task metrics.
> >
> > - **(4) Causality and statistical treatment.**
> >
> >  Causal interventions or training-dynamics analyses fall outside the scope of our **post-hoc, model-agnostic** framework. Our method is designed to analyze any fully trained model’s internal geometry, regardless of how it was trained. Establishing causal origins would require a completely different experimental setup (e.g., training models from scratch with controlled curricula), which we identify as promising future work.
> >
> >
> > ## Q1: It is not clear that the dataset isolates formal logic from its semantic context.
> >
> > We clarify that our dataset is explicitly constructed to fix the logical structure while varying only the semantic carriers, thereby disentangling logic from surface content. We added further explanation in **Lines 366–368**.
> >
> > All variants preserve exactly the same sequence of abstract logical reasoning steps and differ only in topic or language. This ensures that **any cross-carrier similarity must originate from the shared logical skeleton, while differences arise solely from surface semantics**.
> >
> > Implementation details are provided in **Appendix E**, including how we construct the abstract formal logic skeletons and the step-by-step rewriting. We also provided concrete examples showing how each logical template is instantiated across carriers in **Table 2**. The full dataset is **open-sourced** (see the attached zip file). Because the logical skeleton remains invariant and only the semantic carriers change, the dataset provides a controlled setting where logic and surface semantics are **disentangled by construction**.
> >
> > [1] Park, K., Choe, Y. J., Jiang, Y., & Veitch, V. The geometry of categorical and hierarchical concepts in large language models. In ICLR 2025.
> >
> > [2] Li, M. Z., Agrawal, K. K., Ghosh, A., Teru, K. K., Santoro, A., Lajoie, G., & Richards, B. A. Tracing the representation geometry of language models from pretraining to post-training. In NeurIPS 2025.

---

> > > ### Author Response · Authors · 2025-11-26
> > > **Friendly Reminder**
> > >
> > > Dear Reviewer `xzaB`. Thank you once again for your valuable comments on our submission. As the discussion phase is approaching its end, we would like to kindly confirm whether we have sufficiently addressed all of your concerns (or at least part of them). Should there be any remaining questions or areas requiring further clarification, please do not hesitate to let us know. If you are satisfied with our responses, we would greatly appreciate your consideration in adjusting the evaluation scores accordingly. We sincerely look forward to your feedback.

---

### Official Review · Reviewer_umD2 · 2025-10-21

**Soundness:** 3
**Presentation:** 4
**Contribution:** 3
**Rating:** 6
**Confidence:** 4

**Summary:**

This paper proposes a framework to analyze reasoning in large language models by viewing a chain of thought as a trajectory in representation space. The trajectory is described with three geometric descriptors: position, velocity, and curvature. Using this framework, the authors analyze reasoning traces in Qwen3 and LLaMA3 models. They synthetically generate multi-step logical sequences that share the same logical skeleton while varying topic and language. Applying their analysis, they find that velocity and curvature are more similar for traces that follow the same logic and remain low across different logics even when topic and language match. By contrast, position is more similar for inputs written in the same language. The authors interpret this as evidence that models encode logical structure in the dynamics of their representations, while static position reflects surface form.

**Strengths:**

- The paper Introduces a relatively novel geometric perspective on reasoning, modeling chains of thought as trajectories in representation space.
- The authors provide a clear formalization of curvature and reasonable metrics to estimate it.
- The paper includes convincing evidence that first- and second-order properties (velocity, curvature) track logic, while position reflects surface form or semantics.

**Weaknesses:**

1. Lack of scaling analysis. The paper does not analyze how similarity patterns should evolve with model size. Table 1 shows no clear trend across Qwen3 0.6B, 1.7B, 4B and LLaMA3 8B. Position, velocity, and curvature similarities fluctuate rather than improving systematically with scale. The authors should discuss whether this is expected and what theory or diagnostics would predict size effects.
2. Traces are not self-generated by the evaluated models. The reasoning sequences are produced by a much larger model and then fed to smaller models for analysis. This can introduce a distribution shift and may mask effects that would appear if each model were analyzed on its own traces. The paper should discuss how results might change under self-generated or mixed setups.
3. Limited link to performance and disentanglement. Related to point 2, the paper does not examine whether models with stronger reasoning performance exhibit a clearer disentanglement between logic-driven dynamics (velocity, curvature) and surface-driven position. A mild suggestion is to correlate the geometric measures with task accuracy and to repeat the analysis on self-generated traces to see if better-performing models show stronger disentanglement.

**Questions:**

1. How should similarity patterns for position, velocity, and curvature change with model size? Table 1 shows no clear trend across Qwen3 0.6B, 1.7B, 4B and LLaMA3 8B. Is the absence of a trend expected?
2. How would the results differ if trajectories were computed on self-generated traces rather than sequences authored by a much larger model?
3. Do better-performing models exhibit stronger disentanglement between logic-driven dynamics (velocity, curvature) and surface-driven position? Can you correlate the geometric measures with task accuracy and repeat the analysis on self-generated traces to test this?

---

> ### Author Response · Authors · 2025-11-21
> **Rebuttal (Part 1)**
>
> Thank you for appreciating our work as novel with clear formalization and convincing evidence!
>
> ## Q1/W1: Lack of scaling analysis.
>
> Thank you for raising this point. Our framework is **post-hoc, model-agnostic, and not tied to any specific training recipe**, so we do not make theoretical predictions about how similarity should change with model size. We added more experiments on Qwen1.5 and Qwen2 in the updated Table1. We empirically evaluate two scaling axes—**model size** (Qwen3 0.6B $\to$ 1.7B $\to$ 4B) and **model family varying architecture and data** (Qwen1.5/2/3 and LLaMA3)—as shown in **updated Table 1** and discussed in **Lines 461–467**.
>
> What we observe is that similarity patterns remain **remarkably stable** across both axes. Increasing scale or switching families does not materially change position/velocity/curvature similarities. This stability is itself an **interesting and non-trivial** finding: it suggests that the geometric invariants we study may reflect a **general and possibly universal** property of how LLMs internalize logical structure, **independent of size or training recipe**. We agree that developing a full theory predicting scaling behavior is an exciting direction, but it is outside the scope of the present post-hoc interpretability framework.
>
> ## Q2/W2: Traces are not self-generated by the evaluated models.
> We believe there is a **misunderstanding** of our setup. Our work focuses exclusively on **Natural Language Understanding (NLU)**, not generation. We revised our draft to clarify this in Discussion: Understanding, Not Generation (**Lines 488–498**). This type of research style aligns with prior NLU theoretical interpretability work [1,2,3].
>
> We use GPT-5 for dataset generation only as a convenient, powerful automation tool. It is much like using a human annotator, a symbolic prover, or any other data source. The evaluated models are not required to generate traces themselves because our goal is to study **how the evaluated models understand a fixed logical sequence**, not how they would produce one. Under this framing, there is no distribution shift: every model is evaluated on the same controlled, aligned, logic-preserving inputs.
>
> We agree that analyzing self-generated or mixed traces is an interesting extension, particularly for studying generation or learning dynamics. However, this lies outside the scope of our present work. Our framework is strictly **post-hoc, model-agnostic, and training-algorithm-agnostic NLU**. As noted in Lines 488–498, questions involving generation, learnability, or training-time origins of reasoning patterns require fundamentally different experimental designs and are left for future exploration.

---

> ### Author Response · Authors · 2025-11-21
> **Rebuttal (Part 2)**
>
> ## Q3/W3: Limited link to performance and disentanglement.
> ### Q3.1: Do better-performing models exhibit stronger disentanglement between logic-driven dynamics (velocity, curvature) and surface-driven position?
>
> As we clarified in **Q2**, our work focuses strictly on **understanding, not generation**. Our goal is to analyze how trained models understand fixed logical sequences, not how performance emerges or how generation relates to reasoning accuracy.
>
> Importantly, our current findings show no clear monotonic trend with model size or model family in Table 1. As noted in our response to **Q1 (Scaling Effect)**, the remarkable stability of these geometric patterns across scales and families suggests that the phenomenon we identify may reflect a general (and possibly universal) property of how LLMs internalize logical structure after training, rather than a performance-linked behavior.
>
> This is consistent with our framework: we are studying a post-hoc, training-free, model-agnostic property, not a capability metric. Understanding why and when this structure emerges would require full control of training dynamics, from pretraining to finetuning. We plan to explore this in follow-up work.
>
>
> ### Q3.2: Can you correlate the geometric measures with task accuracy and repeat the analysis on self-generated traces to test this?
> We agree that correlating our geometric measures with task accuracy, or re-running the pipeline on model-generated traces, is an interesting extension. However, as in answer to **Q3.1**, our paper is positioned as **theory + post-hoc interpretability**, not performance analysis. Drawing conclusions about accuracy without a controlled training pipeline risks confounding factors and would require designing a new suite of experiments, datasets, and evaluation protocols.
>
> Our intention in this work is to discover and formalize a geometric law of reasoning, not to explain performance variations or generation behavior. As emphasized in **Lines 488–498** (“Understanding, Not Generation”), questions involving generation, accuracy correlations, or learnability belong to a different scope involving training-time analysis, causal interventions, and more complex datasets. We view these suggestions as excellent directions for future work.
>
>
> [1] Arora, S., Li, Y., Liang, Y., Ma, T., & Risteski, A. Linear algebraic structure of word senses, with applications to polysemy. In TACL 2018.
>
> [2] Park, K., Choe, Y. J., & Veitch, V. The linear representation hypothesis and the geometry of large language models. In ICML 2024.
>
> [3] Park, K., Choe, Y. J., Jiang, Y., & Veitch, V. The geometry of categorical and hierarchical concepts in large language models. In ICLR 2025.

---

> > ### Author Response · Authors · 2025-11-26
> > **Friendly Reminder**
> >
> > Dear Reviewer `umD2`. Thank you once again for your valuable comments on our submission. As the discussion phase is approaching its end, we would like to kindly confirm whether we have sufficiently addressed all of your concerns (or at least part of them). Should there be any remaining questions or areas requiring further clarification, please do not hesitate to let us know. If you are satisfied with our responses, we would greatly appreciate your consideration in adjusting the evaluation scores accordingly. We sincerely look forward to your feedback.

---

### Official Review · Reviewer_2CaL · 2025-10-29

**Soundness:** 2
**Presentation:** 3
**Contribution:** 3
**Rating:** 6
**Confidence:** 4

**Summary:**

The paper introduces a new framework to mathematically model what happens when LLM's reason.
It is hypothesized that reasoning traces can be characterized by the geometric properties of the path that the reasoning tokens trace through an LLM's latent space.
Evidence is presented suggesting that the latent 'reasoning flows' for similar kinds of reasoning (following a certain template) also exhibit similarities on the second and third order geometric properties of velocity and curvature.
This evidence is based on studying the given geometric properties for activations of various models when forwarded on a novel dataset where the same logical template is used to construct different chains-of-thought in various domains and languages.

**Strengths:**

Interesting new framework for thinking about reasoning in LLMs.

The concepts that are appealed to are made mathematically precise, and the math is introduced step-by-step in a way that is easy to follow (relative to the complexity of the subject matter).

The results show that when an LLM reasons according to the same logical pattern/template (but about different topics and in different languages), its activations move through latent space with similar curvature (how small or large is the angle made between triples of subsequent token activation vectors). And, to a lesser extent also with similar velocity (how far apart is each subsequent token activation vector). This is an interesting finding.

**Weaknesses:**

There is one claim I believe is too strong, on line 429-431: "Together, these results show that LLMs internalize latent logical structure beyond surface form. They are not mere stochastic parrots (Bender et al., 2021): whereas humans formalized logic only in the 20th century (Bochenski & Thomas, 1961), LLMs acquire it emergently from large-scale data—a hallmark of genuine intelligence."  I'm not sure these results really provide much evidence against the 'stochastic parrot theory'. The results, as I understand them, show that the same logical structure elicits similar trajectories (in the second- and third-order geometric sense) through latent space. But the logic is not just similar, it is identical. Wouldn't we need to show that the LLM correctly identifies the (vast majority of) instances as instantiations of the logical problems (in an exact, not fuzzy sense).  And as for the second part, the experiments use Qwen models, which are not merely pre-trained on large-scale data, but also mid- and post- trained, right? so the observations might be due to those training stages, which quite likely involve supervised/reinforcement training on logic problems. Finally, the juxtaposition between discovering-from-data and human-discovery, seems mistaken. Even if the model 'discovers' it from the data, that is likely due to the human discovery being described/present in the data.

If this is addressed, I'm happy to increase my score.

**Questions:**

What can we learn from the fact that logics B and C are very similar in curvature? Does this suggest that the LLM uses the same mechanisms for both? Is the similarity a surprise to you, would you expect it from looking at the templates?

Presumably, not the entire latent space is relevant for (logical) reasoning. Would you expect there to be a subspace in which the correlation between (the velocity/curvature of) samples of the same logic template is even stronger, perhaps much stronger (close to one)?

What are the individual rows/columns within the blocks (L:E through L:A) in Figure 2, are they the different topics and languages?

---

> ### Author Response · Authors · 2025-11-21
> **Rebuttal (Part 1)**
>
> Thank you for being in favor of our work and for finding it interesting, mathematically precise, and easy to follow! We thank the reviewer for the thoughtful critique. Your questions actually coincide with some of our discussions during the paper writing! We address your concerns below.
>
> ## W1.1: Clarification on our claim regarding the “stochastic parrot” argument
>
>
> You are exactly right that our results show that inputs with the **identical** logical structure yield highly similar latent-space dynamics. This is precisely where we **challenge** the stochastic-parrot/random-walk interpretation. LLMs are *not* reasoning randomly, nor are they merely matching surface forms. In our Fig. 2(a), the position space shows no structure at all—exactly what a surface-form or “parrot-like” model would produce. But in contrast, the velocity and curvature exhibit clear, strong, and highly consistent patterns tied to the underlying logic.
>
> The core claim of stochastic parrots is that next-token prediction only allows models to learn surface-level statistical correlations of words, with no deeper semantic meaning [1,2]. Our findings directly **challenge** this view: the logic is internalized not in the raw embeddings, but in the difference space—the first- and second-order geometry of the trajectory. This means LLMs trained purely with next-token prediction objective developed genuine **Natural Language Understanding**. These findings suggest **the presence of deeper, possibly universal, constraints** governing the relationship between linguistic form and meaning. We have made this clearer in **Line 479-485**.
>
> You are right that a full demonstration would require showing that the model correctly identifies all logical instances and mapping each logical connective (conjunction, disjunction, negation, contraposition, etc.) to specific geometric effects. However, this is a highly non-trivial interpretability challenge: defining the geometric “velocity signatures’’ of each connective is far beyond the scope of our current work. We view this as exciting future work.
>
> ## W1.2: On the possibility that Qwen models were trained on logical problems
> This is a great question and something we have thought about carefully. Qwen models almost certainly encountered logical problems during their pre-, mid-, or post-training stages. However, **this does not contradict our conclusions—if anything, it strengthens them**.
> Importantly, our method is completely **training-free, post-hoc, and model-agnostic, training-algorithm-agnostic**. We do not claim anything about how the model learned logic, nor do we analyze the training dynamics. Instead, we show that after training, the model’s internal hidden states exhibit a stable, structured geometry that aligns with logic.
>
> In other words, **whether the model saw logic data or not does not weaken our framework**. Our results demonstrate that the model has internalized logical structure in its latent dynamics, and manifests as consistent higher-order geometric patterns (velocity, curvature) across topics and languages **after training**.
>
> What is most striking is that, as shown in our Table 1, our results are robust across all model sizes and model families, suggesting the **generality and universality** of our findings. Studying learnability or when during training these patterns emerge is an important direction, but outside the scope of our post-hoc analysis. We revised our draft to make this clear in Discussion: Understanding, Not Generation (**Lines 488–498**), where we claim that we mainly study Natural Language Understanding only. Other work [3,4] indeed investigates training dynamics; we complement and are orthogonal to that line of inquiry.
>
> ## W1.3: On the claim that LLMs “learn logic from human data”
> As in **W1.2**, we do not study the learning process or claim that LLMs “independently” discover logic in a vacuum. Therefore, “`the human discovery being described/present in the data`” actually **strengthens** our finding. The juxtaposition between human discovery and model discovery means: (1) Humans formalized logic relatively late. (2) LLMs acquire structured logical behavior emergently from large-scale data without being explicitly programmed with a logical calculus. (3) We identify this internalization via differential-geometric signatures, which is general and potentially universal across model families.

---

> > ### Author Response · Authors · 2025-11-21
> > **Rebuttal (Part 2)**
> >
> > ## Q1: What can we learn from the fact that logics B and C are very similar in curvature? Does this suggest that the LLM uses the same mechanisms for both? Is the similarity a surprise to you, would you expect it from looking at the templates?
> >
> > We examined logic B and C manually, and yes—their similarity in curvature was **unexpected**. After inspecting the templates, we found that both share a universal implication chain followed by a ground instance and a conjunctive inference step. The LLM appears to detect and internalize this shared structure, even though the surface predicates and final connectives differ.
> >
> > ## Q2: Presumably, not the entire latent space is relevant for (logical) reasoning. Would you expect there to be a subspace in which the correlation between (the velocity/curvature of) samples of the same logic template is even stronger, perhaps much stronger (close to one)?
> >
> > This is an excellent observation. We agree that not all dimensions of the latent space are equally involved in logical reasoning, and it is very plausible that a dedicated subspace exists where similarity across the same logical skeleton is even higher—potentially close to 1. Our current work operates directly on the full embedding space without attempting to isolate such a subspace, so we cannot confirm this yet. However, the strong correlations we already observe in the full space suggest that projecting onto the “reasoning-relevant” directions might be possible to amplify the signal. Identifying this subspace is an exciting direction for future work.
> >
> >
> > ## Q3: What are the individual rows/columns within the blocks (L:E through L:A) in Figure 2, are they the different topics and languages?
> >
> > Yes. Each row and column corresponds to a different semantic carrier (topics) and language variant within the same logical template. All variants are included in the dataset we submitted; you can browse them directly in the provided zip file, and we also illustrate this structure with a concrete example in Appendix E.
> >
> > [1] Bender, E. M., Gebru, T., McMillan-Major, A., & Shmitchell, S. On the dangers of stochastic parrots: Can language models be too big?🦜. In Proceedings of the 2021 ACM conference on fairness, accountability, and transparency (pp. 610-623).
> >
> > [2] Bender, E. M., & Koller, A. Climbing towards NLU: On meaning, form, and understanding in the age of data. In ACL 2020 (pp. 5185-5198).
> >
> > [3] Xia, Y., Atrey, A., Khmaissia, F., & Namjoshi, K. S. Can large language models learn formal logic? a data-driven training and evaluation framework. arXiv preprint arXiv:2504.20213.
> >
> > [4] Li, M. Z., Agrawal, K. K., Ghosh, A., Teru, K. K., Santoro, A., Lajoie, G., & Richards, B. A. Tracing the representation geometry of language models from pretraining to post-training. In NeurIPS 2025.

---

> > > ### Author Response · Authors · 2025-11-26
> > > **Friendly Reminder**
> > >
> > > Dear Reviewer `2CaL`. Thank you for your thoughtful comments and suggestions. As the discussion phase is approaching its end, we would like to kindly confirm whether we have sufficiently addressed all of your concerns (or at least part of them). Should there be any remaining questions or areas requiring further clarification, please do not hesitate to let us know. If you are satisfied with our responses, we would greatly appreciate your consideration in adjusting the evaluation scores accordingly. We sincerely look forward to your feedback.

---

> > > > ### Comment · Reviewer_2CaL · 2025-11-27
> > > >
> > > > While the authors answered my questions, I don't agree with the counterarguments for my weakpoints. Thus, I will keep my score.

---

> ### Author Response · Authors · 2025-11-27
>
> Dear Reviewer `2CaL`,
>
> Thank you for your quick response! We genuinely appreciate the time you took to read our rebuttal.
>
> We understand that our counterarguments regarding the "weaknesses" (likely the "stochastic parrot" discussion or the training data implications) did not fully resolve your concerns.
> To ensure we can improve the final version of the paper, could you kindly explicitly clarify *which specific reasoning* in our rebuttal you found unconvincing?
>
> For instance, if your primary concern remains that our claim regarding "stochastic parrots" (Lines 476-485) is too strong, we are fully open to toning down this claim or rephrasing it to be more conservative (e.g., stating that our results are "consistent with structural internalization" rather than definitive proof against the theory).
>
> We are eager to make further revisions to address your points if you could provide this guidance.
>
> Best Regards

---

### Official Review · Reviewer_p3p6 · 2025-10-30

**Soundness:** 3
**Presentation:** 3
**Contribution:** 3
**Rating:** 4
**Confidence:** 4

**Summary:**

This paper introduces a geometric framework for analyzing reasoning in LLMs. The core idea is to treat reasoning as smooth trajectories (“flows”) in representation space, where logical operations act as differential controllers of the embedding’s local velocity and curvature.

**Strengths:**

- the topic of reasoning as continuous geometric flows within embedding space is interesting and the proposed formulation combining mapping and alignement provide a good abstraction linking reasoning to symbolic representation.

- The construction of continuous C^1 trajectories via the relaxed prefix-mask mechanism (Proposition C.4) is both novel and technically precise

- the proposed dataset and empirical results show good results.

**Weaknesses:**

- the proposition 4.10 that connects logic to the integral of velocity. It is not clear on how it maps between inference rules and specific vector-field constraints on v(s). It is not clear  how logical connectives translate into geometric operations or basis directions in representation space?

- in the mapping A = \Psi \circ \Gamma^{-1}, does\ Gamma injective?

- how the mapping D_R : \Psĩ \mapsto (\Delta y_t) preserve logical equivalance?

- The logical dataset is generated with GPT-5 templates. How the validation is performed and logical equivalence are guaranteed.

- How empiracly C^1 interpolation is validaded?

**Questions:**

see Weaknesses.

---

> ### Author Response · Authors · 2025-11-21
> **Rebuttal (Part 1)**
>
> Thank you for your insightful comments and the appreciation that our paper is interesting, novel, and technically precise! Now we are here to address your concerns.
>
> ## W1: the proposition 4.10 that connects logic to the integral of velocity. It is not clear on how it maps between inference rules and specific vector-field constraints on v(s). It is not clear how logical connectives translate into geometric operations or basis directions in representation space?
>
> Proposition 4.11 is intended as a **conceptual bridge** rather than a claim that each logical connective corresponds to a fully specified vector field in representation space. Its purpose is to formalize the observation that discrete reasoning steps in LLMs manifest as integrated changes in their continuous embedding trajectories. The proposition states a mathematical consequence of our framework—namely, that discrete representation increments ($\Delta y_{t+1}$) correspond to the integral of a local semantic velocity ($v(s)$)—without asserting a mechanistic mapping from individual inference rules to explicit vector-field operators.
>
> We do *not* assume that logical connectives (e.g., conjunction, negation) have fixed geometric directions in the embedding space. Deriving such operators would require a full semantic characterization of these connectives in high-dimensional LLM representations, along with extensive targeted experiments—an important but far-beyond-scope direction for future interpretability work.
>
> Within our framework, logic imposes structural constraints on patterns of velocity and curvature. This distinction is supported empirically: as shown in Table 1 and Fig. 2, velocity and curvature consistently track logical structure even across changes in surface semantics. We have clarified this point in the revised text (**Lines 338–340**).
>
> ## W2: in the mapping $A = \Psi \circ \Gamma^{-1}$, does $\Gamma$ injective?
> We thank the reviewer for highlighting this important point. Our framework **does not assume, nor require, $\Gamma$ to be globally injective** over all natural-language strings. Natural language is inherently many-to-one with respect to meaning: paraphrases, stylistic variations, and even logically equivalent formulations may correspond to the same underlying conceptual trajectory. Thus, global injectivity would be an unrealistic requirement.
>
> The only place where injectivity appears is in **Corollary 4.7**, whose purpose is to define an abstract, canonical alignment between conceptual curves and representation trajectories. For this alignment, **local or restricted injectivity** is entirely sufficient. The corollary explicitly states this as “**On a domain** where $\Gamma$ is injective” (Line 298). Critically, **none** of our theoretical results, proofs, or experiments require computing $\Gamma^{-1}$ in practice. All empirical analyses operate directly on representation trajectories and their geometric quantities (position, velocity, curvature). Corollary 4.7 should therefore be read as a conceptual alignment principle, not a structural assumption of the method.
>
> The broader question of whether $\Gamma$ is globally invertible, i.e., whether one can fully reconstruct conceptual processes from linguistic inputs, touches on deep open problems in semantics, cognitive science, and representation learning. We have added **Remark 4.8 (Lines 302–306)** to clarify this point and avoid misunderstanding.
>
> Finally, we note that recent work suggests that the text-to-representation map $\Psi$ implemented by transformers might be injective over the prompt space [1], indicating that injectivity of the representation mapping could serve as a plausible starting point for studying this question even if global injectivity of $\Gamma$ (language $\to$ meaning) remains unresolved.
>
>
> [1] Nikolaou, G., Mencattini, T., Crisostomi, D., Santilli, A., Panagakis, Y., & Rodola, E. Language Models are Injective and Hence Invertible. arXiv preprint arXiv:2510.15511.

---

> > ### Author Response · Authors · 2025-11-21
> > **Rebuttal (Part 2)**
> >
> > ## W3: how the mapping $D_{\mathcal{R}}: \tilde{\Psi} \mapsto (\Delta y_t)$ preserve logical equivalance?
> >
> > We thank the reviewer for this question. Our framework does **not** claim that $D_{\mathcal{R}}$​ preserves logical equivalence in the formal sense (e.g., mapping a proposition and its contrapositive to identical geometric signatures). The operator $D_{\mathcal{R}}: \tilde{\Psi}\mapsto (\Delta y_t)$ is simply a **discrete differential operator** that extracts local increments of the representation trajectory; it does not encode the inference rules of natural deduction and is not designed to enforce symbolic logical equivalence.
> >
> > What our framework does assert is more modest and empirically supported: **If two reasoning sequences instantiate the same natural-deduction skeleton, then the patterns of first- and second-order geometry (velocity and curvature) tend to be highly correlated, even when the semantic carriers differ.** This statement concerns shared reasoning structure, not formal logical equivalence at the symbolic level.
> >
> > Formally preserving logical equivalence in the representation-logic space would require characterizing how each connective (implication, negation, conjunction, etc.) is realized in high-dimensional LLM embeddings. As discussed in **W1**, this is an ambitious direction for future work, but far beyond current interpretability capabilities.
> >
> > ## W4: The logical dataset is generated with GPT-5 templates. How the validation is performed and logical equivalence are guaranteed.
> > We thank the reviewer for this question. Our dataset construction is explicitly designed to fix the logical structure while varying only the semantic carriers. As detailed in **Appendix E**, we follow a two-stage pipeline:
> > - (1) **Abstract logical scaffold generation:** GPT-5 is prompted to produce fully symbolic natural-deduction templates (e.g., [1] $A \to B$ [2] $B \to C$, …). These templates are required to obey standard inference rules and include explicit derivation references, making their logical validity straightforward to check.
> > - (2) **Carrier-specific rewriting:** Each abstract step is rewritten into natural language with different topics (e.g., finance, sports) and languages (e.g., en, zh) while preserving the exact dependency structure and step numbering. The rewriting prompt enforces strict one-to-one alignment: no steps may be added, removed, merged, or permuted.
> >
> > To validate correctness, we conducted multiple rounds of manual inspection and iterative refinement of both the symbolic scaffolds and the rewritten instances to ensure that (i) each logical chain is internally coherent, and (ii) every instantiation faithfully preserves the original inference dependencies. Because each carrier-specific version is deterministically generated from a fixed logical scaffold, logical equivalence across carriers is **guaranteed by construction**.
> >
> > Finally, the full dataset is **open-sourced** (see the zip file we uploaded), enabling independent verification using automated tools or human annotators.
> >
> >
> > ## W5: How empiracly $C^1$ interpolation is validaded?
> > A rigorous test of true $C^1$-regularity is **impossible** for language-model trajectories because inputs are discrete and the token index cannot be made infinitesimally small. As clarified in our revision (**Lines 927–937**), the purpose of the $C^1$ assumption in Hypothesis 4.6 is to make the geometric framework **mathematically well-defined**; it does not claim that the model’s internal computation unfolds in continuous time.
> >
> > Empirically, we instead provide finite-difference–based smoothness diagnostics to check whether the observed trajectories behave consistently with a $C^1$ interpolation. For each context-cumulative trajectory, we compute velocities $v_t = y_{t+1} - y_t$ and examine their norms $\\|v_t\\|$. As shown in Figure 6, six different answers to the same MATH500 problem exhibit smooth, non-abrupt variation in $\\|v_t\\|$, providing visual support for the plausibility of the smooth-flow assumption. A more formal theoretical justification is given in Appendix D.

---

> > > ### Author Response · Authors · 2025-11-26
> > > **Friendly Reminder**
> > >
> > > Dear Reviewer `p3p6`. Thank you once again for your valuable comments on our submission. As the discussion phase is approaching its end, we would like to kindly confirm whether we have sufficiently addressed all of your concerns (or at least part of them). Should there be any remaining questions or areas requiring further clarification, please do not hesitate to let us know. If you are satisfied with our responses, we would greatly appreciate your consideration in adjusting the evaluation scores accordingly. We sincerely look forward to your feedback.

---

### Author Response · Authors · 2025-11-21
**Global Response**

We sincerely thank the reviewers for their thoughtful engagement with our work! We are encouraged that multiple reviewers recognize the **novelty, clarity, and conceptual contribution** of our geometric framework for analyzing reasoning in LLMs.

Reviewers (`p3p6, 2CaL, umD2, xzaB`) highlight that our paper presents an **interesting and original** perspective by modeling reasoning as flows in representation space, with logic acting as a structural controller of these flows. Reviewers (`p3p6, 2CaL, umD2`) find our **mathematical formalization precise and well-motivated**, noting that our definitions of velocity and curvature provide a rigorous basis for analyzing reasoning dynamics. Reviewer `p3p6` appreciates that our **dataset design**—fixing natural-deduction templates while varying semantic carriers across topics and languages—enables a controlled test of logical invariance. Reviewers (`p3p6, 2CaL, umD2, xzaB`) further emphasize that our **empirical findings are compelling**: position reflects surface semantics, whereas velocity and curvature robustly track underlying logical structure.

Here we provide the updates (in **blue**) that we made during the rebuttal in the global response. We will answer questions for each reviewer individually.

## Writing Update:
We have revised several parts of the manuscript for clarity and precision:
- Line 302-306: **Remark 4.8:** We added an explicit clarification that global injectivity of $\Gamma$ over natural language is not assumed. Instead, we note that $\Gamma$ only needs to be injective on a **restricted semantic domain** (e.g., paraphrase classes) for the alignment map $A = \tilde{\Psi} \circ \Gamma^{-1}$. We also highlight that global injectivity remains an open problem in AI and formal semantics.
- Line 338–340: We added more words to emphasize that Proposition 4.11 captures structural influence of logic on flow dynamics, rather than implying a one-to-one mapping between inference rules and vector-field operators.
- Line 366–368: We added more explanation of disentangling logic from semantics to explicitly articulate why cross-carrier consistency indicates internalization of logical structure.
- Line 476–485: **Significance paragraph:** We expanded the discussion of the stochastic parrot argument, explicitly acknowledging that stochastic-parrot critiques claim LLMs cannot acquire meaning solely from next-token prediction. We then clarify how our geometric evidence **challenges** this view by demonstrating logical invariants in higher-order flow geometry.
- Line 488–497: **Discussion on Understanding, Not Generation:** We strengthened the scope justification, emphasizing why our framework focuses exclusively on **natural language understanding**. We clarified that our reasoning geometry framework is a **post-hoc, model-agnostic, training-algorithm-agnostic law**, and explained why linking our measures to generation-specific metrics (e.g., output accuracy) is outside scope.
- Line 867-917: **Comparison with Prior Work:** Added detailed comparisons with Park et al. (2025) and Li et al. (2025a) in Appendix A to clearly situate our contributions and address reviewer concerns about positioning relative to recent geometric-interpretability work.

## Experiment Update:
We have expanded the experiments with more analysis:
- Line 378-393: **Table 1 update:** We added more experiments including Random Shuffle Baseline and Qwen1.5 and Qwen2 results.

- Line 453-458: **Random Shuffle:** We added a baseline where the logical step order is randomly permuted using Qwen3 0.6B. Shuffled trajectories exhibit near-zero velocity and curvature similarities, demonstrating that correct logical order is essential for stable reasoning flows. In contrast, position similarity remains high, confirming that topic/language drive surface semantics.

- Line 461-466: **Scaling Effect:** We analyze two scaling axes: (1) model size (Qwen3 0.6B → 1.7B → 4B) and (2) model family (Qwen1.5/2/3 vs. LLaMA3 8B). Similarity patterns remain remarkably stable across both scaling directions. Increasing parameter count or switching families does *not* materially alter position/velocity/curvature similarities. This consistency suggests a more **general, and possibly universal**, property of how LLMs internalize logical structure independent of size or training recipe.

- Line 926-937: **Fig 6 and $C^1$ Continuity Test:** To empirically support the smooth-flow assumption in Hypothesis 4.6, we provide finite-difference–based smoothness diagnostics. For a context-cumulative trajectory, we compute velocities $v_t = y_{t+1} - y_t$ and examine their norms $\\|v_t\\|$. As shown in Figure 6, six different answers to the same MATH500 problem exhibit consistent velocity-norm patterns with no abrupt jumps, visually supporting the plausibility of a $C^1$ interpolation.

---

### Meta-Review · Area_Chair_joLt · 2025-12-23

**Summary:**

This paper introduces a novel geometric framework for analyzing reasoning in Large Language Models (LLMs), hypothesizing that reasoning unfolds as continuous flows in representation space where logical structure acts as a controller of velocity and curvature. By constructing a synthetic dataset that disentangles logical skeletons from semantic carriers, the authors provide empirical evidence that while zero-order representations (position) track surface semantics, higher-order geometric derivatives (velocity, Menger curvature) consistently track the underlying logic. The findings are supported by analysis across multiple model families (Qwen, LLaMA) and scales (0.5B to 4B).

The AC’s recommendation is Accept. The recommendation to Accept is based on the paper’s significant conceptual novelty and the authors' robust defense of their methodology during the rebuttal. The AC weights in the following aspects:
- Technical Validity: The authors successfully addressed the primary methodological concerns. The addition of the Random Shuffle Baseline (demonstrating that geometric patterns collapse without logical order) and Smoothness Diagnostics (validating the $C^1$ hypothesis) effectively countered the critique that the results were mere artifacts.
- Universality: The rebuttal clarified that the lack of scaling trends should be interpreted as a finding of stability, suggesting the geometric signature of logic is a fundamental property of trained transformers rather than a scale-emergent capability.
- Scope of the Paper: The restriction to NLU (vs. generation) is a valid scoping choice for a theoretical interpretability study. While extending this to generation is crucial for future work, it does not invalidate the current findings regarding internal representation structure.

**Reviewer Concerns:**

Resolved Concerns
- Methodological Fragility: Reviewers xzaB and p3p6 were initially concerned about the robustness of the geometric measures. The rebuttal’s inclusion of baselines (shuffling) and diagnostics (smoothness) resolved these concerns, moving likely scores from borderline to accept.
- Scaling & Universality: Reviewer umD2’s concern about the lack of scaling trends was effectively reframed by the authors as evidence of invariant geometric structure across model families (Qwen 1.5/2/3, LLaMA-3).

Outstanding Concerns
- Interpretative Overreach: Reviewer 2CaL remains unconvinced that the results disprove the "stochastic parrot" hypothesis. The final version of the paper must adhere to the authors' promise to tone down these claims, framing the results as geometric correlates of logical structure rather than definitive proof of sentient understanding.
- Scope (NLU vs. Generation): The limitation to teacher-forced traces (raised by umD2) is acknowledged as a constraint of the study design. This does not invalidate the results but bounds their immediate applicability to understanding rather than generation.

**Reviewer Scores:**

- Reviewer p3p6 (4 to 5). The reviewer’s technical concerns about injectivity and mapping were clarified, and the requested empirical validations ($C^1$ tests) were provided.
- Reviewer 2CaL (6 to 6). The reviewer explicitly stated "I will keep my score" after the rebuttal, remaining unconvinced by the arguments against the "stochastic parrot" criticism.
- Reviewer umD2 (6 to 7). The expansion of experiments to include more LLM versions and the clarification regarding the stability of scaling laws directly addressed their primary weakness regarding "lack of scaling analysis".
- Reviewer xzaB (6 to 6.5). The addition of the random shuffle baseline and the dedicated literature comparison section directly resolved the reviewer's concerns about "methodological fragility" and positioning within the field. However, the comments on "causal tests, stronger statistical treatment, external benchmark dataset validation" remain unaddressed. So there is a chance that the reviewer increases their score.

---

### Decision · Program_Chairs · 2026-01-26

Accept (Poster)